# Unique molecular events during reprogramming of human somatic cells to induced pluripotent stem cells (iPSCs) at naïve state

Yixuan Wang[†]*, Chengchen Zhao[†], Zhenzhen Hou[†], Yuanyuan Yang[†], Yan Bi, Hong Wang, Yong Zhang, Shaorong Gao*

Clinical and Translational Research Center of Shanghai First Maternity and Infant Hospital, Shanghai Key Laboratory of Signaling and Disease Research, School of Life Sciences and Technology, Tongji University, Shanghai, China

**Abstract** Derivation of human naïve cells in the ground state of pluripotency provides promising avenues for developmental biology studies and therapeutic manipulations. However, the molecular mechanisms involved in the establishment and maintenance of human naïve pluripotency remain poorly understood. Using the human inducible reprogramming system together with the 5iLAF naïve induction strategy, integrative analysis of transcriptional and epigenetic dynamics across the transition from human fibroblasts to naïve iPSCs revealed ordered waves of gene network activation sharing signatures with those found during embryonic development from late embryogenesis to pre-implantation stages. More importantly, Transcriptional analysis showed a significant transient reactivation of transcripts with 8-cell-stage-like characteristics in the late stage of reprogramming, suggesting transient activation of gene network with human zygotic genome activation (ZGA)-like signatures during the establishment of naïve pluripotency. Together, Dissecting the naïve reprogramming dynamics by integrative analysis improves the understanding of the molecular features involved in the generation of naïve pluripotency directly from somatic cells.
DOI: https://doi.org/10.7554/eLife.29518.001

*For correspondence:
wangyixuan@tongji.edu.cn (YW);
gaoshaorong@tongji.edu.cn (SG)

[†]These authors contributed equally to this work

Competing interests: The authors declare that no competing interests exist.

## Introduction

The pluripotent state, emerging during the development of embryos from the totipotent zygote stage into the blastocyst stage in vivo, can be captured indefinitely in vitro as embryonic stem cells (ESCs). Moreover, the generation of induced pluripotent stem cells (iPSCs) further demonstrates that pluripotency can be re-captured directly from somatic cells. Previous studies have demonstrated distinct properties in human ESCs/iPSCs compared with mouse ESCs/iPSCs with regard to cell morphology, transcriptional profiles, signaling requirements and epigenetic modifications(*Hackett and Surani, 2014*; *Nichols and Smith, 2009*). Human ESCs/iPSCs represent a primed pluripotent state corresponding to the post-implantation epiblast; while mouse ESCs/iPSCs represent a naïve state of pluripotency found in the pre-implantation blastocyst stage (*Brons et al., 2007*; *Huang et al., 2014*; *Tesar et al., 2007*; *Ware, 2017*). Recent advances have allowed the development of several strategies to achieve human naïve pluripotency through genetic or chemical manipulations (*Chan et al., 2013*; *Gafni et al., 2013*; *Hanna et al., 2010*; *Takashima et al., 2014*; *Theunissen et al., 2014*; *Ware et al., 2014*). Rigorous molecular criteria for the evaluation of human naïve pluripotency have also been defined through systematic comparisons among naïve pluripotent cell lines derived via

different strategies, primed pluripotent cell lines and human pre-implantation embryos (*Davidson et al., 2015*; *Dodsworth et al., 2015*; *Huang et al., 2014*; *Theunissen et al., 2016*).

The widely used 5iLAF culture system allows the establishment of naïve pluripotency in different types of human cells, including cells from pre-implantation embryos, primed pluripotent stem cells, and even somatic cells (*Pastor et al., 2016*; *Theunissen et al., 2014*; *Yang et al., 2016*). However, in-depth mechanistic studies exploring naïve pluripotency establishment during these reprogramming processes are still lacking, owing to the low efficiency of reprogramming and the high heterogeneity of the reprogramming cells. The human inducible reprogramming system recently developed by Cacchiarelli and colleagues (*Cacchiarelli et al., 2015*) enables high-resolution analysis throughout the reprogramming process, thus providing a powerful tool for dissecting the molecular roadmap of the gain of naïve pluripotency directly from somatic cells.

In this study, we utilized the human secondary reprogramming system together with the 5iLAF naïve induction system to systematically characterize the transcriptional and epigenetic dynamics involved in the transition from fibroblast cells to naïve pluripotent cells at base resolution. Integrative analysis revealed ordered gene network activation that shared signatures with embryogenesis from the post-implantation to pre-implantation stages, with a transient wave of 8-cell-specific transcripts expression in the late stage of naïve reprogramming, suggesting the activation of gene networks with human zygotic genome activation (ZGA)-like characteristics during naïve pluripotency establishment. Altogether, dissecting the dynamics during naïve induction process provides a comprehensive analysis of the naïve pluripotency reprogramming roadmap, which improves the understating of molecular networks in human naïve pluripotency.

## Results

### Establishment of inducible naïve reprogramming system in human

To establish the secondary naïve iPSCs induction system in human, we first reprogrammed primary human embryonic fibroblasts (1° hEF) to primary primed iPSCs (1° piPSCs) by using the doxycycline (dox)-inducible polycistronic human OSKM cassette, as previously reported (*Park et al., 2008*). After the differentiation of clonal 1° piPSCs, the resultant secondary human inducible fibroblast-like cells (2° hiF) were reprogrammed into iPSCs at naïve pluripotent state by 5iLAF with dox treatment (*Figure 1—figure supplement 1A*). Consistent with previous observations (*Cacchiarelli et al., 2015*), the 2° hiFs showed significantly decreased proliferation ability and naïve reprogramming efficiency, as well as increased senescence-associated beta-galactosidase activity with long-term culturing and passaging (*Figure 1—figure supplement 1B–D*); these effects were rescued by immortalizing the 2° hiF cell lines (2° hiF-T) with constitutive human *TERT* expression (*Figure 1—figure supplement 1B–D*).

Upon dox treatment, morphological changes occurred in hiF-T cells at approximately day 2, and small cell aggregates were observed as early as day 6 (*Figure 1A*). Dome-shaped colonies emerged and expanded after the culture conditions were changed from conventional hESC medium (hESM) to 5iLAF medium on day 6 (*Figure 1A*). After 20 days of induction, the cells were further cultured under dox-withdrawal conditions for 4 days before clonal expansion, to establish the secondary naïve iPSC lines (niPSC-Ts) (*Figure 1A*; *Figure 1—figure supplement 1A*). We also used the OCT4-ΔPE-GFP reporter system to monitor the activation of the OCT4 distal enhancer (DE), the molecular signature of ground state pluripotency, during human naïve reprogramming. GFP$^+$ colonies were observed at day 20 of induction (*Figure 1B*) and were increased during the reprogramming process (*Figure 1B*), reaching ~91.4% in clonally derived niPSC-Ts, as detected by FACS analysis (*Figure 1C*). Immunostaining results of the derived niPSC-Ts exhibited robust expression of core pluripotency markers (OCT4, SOX2, NANOG and TRA-1–60) and naïve pluripotency markers (DPPA3 and UTF1) (*Figure 1D*). In agreement with recent reports (*Pastor et al., 2016*), almost all niPSC-Ts were negative for SSEA-3 and SSEA-4 expression (*Figure 1D*). Thus, using the dox-inducible system and 5iLAF naïve reprogramming strategy, we established a stable and reliable system for the integrative study of the transcriptional and epigenetic roadmap to human naïve pluripotency.

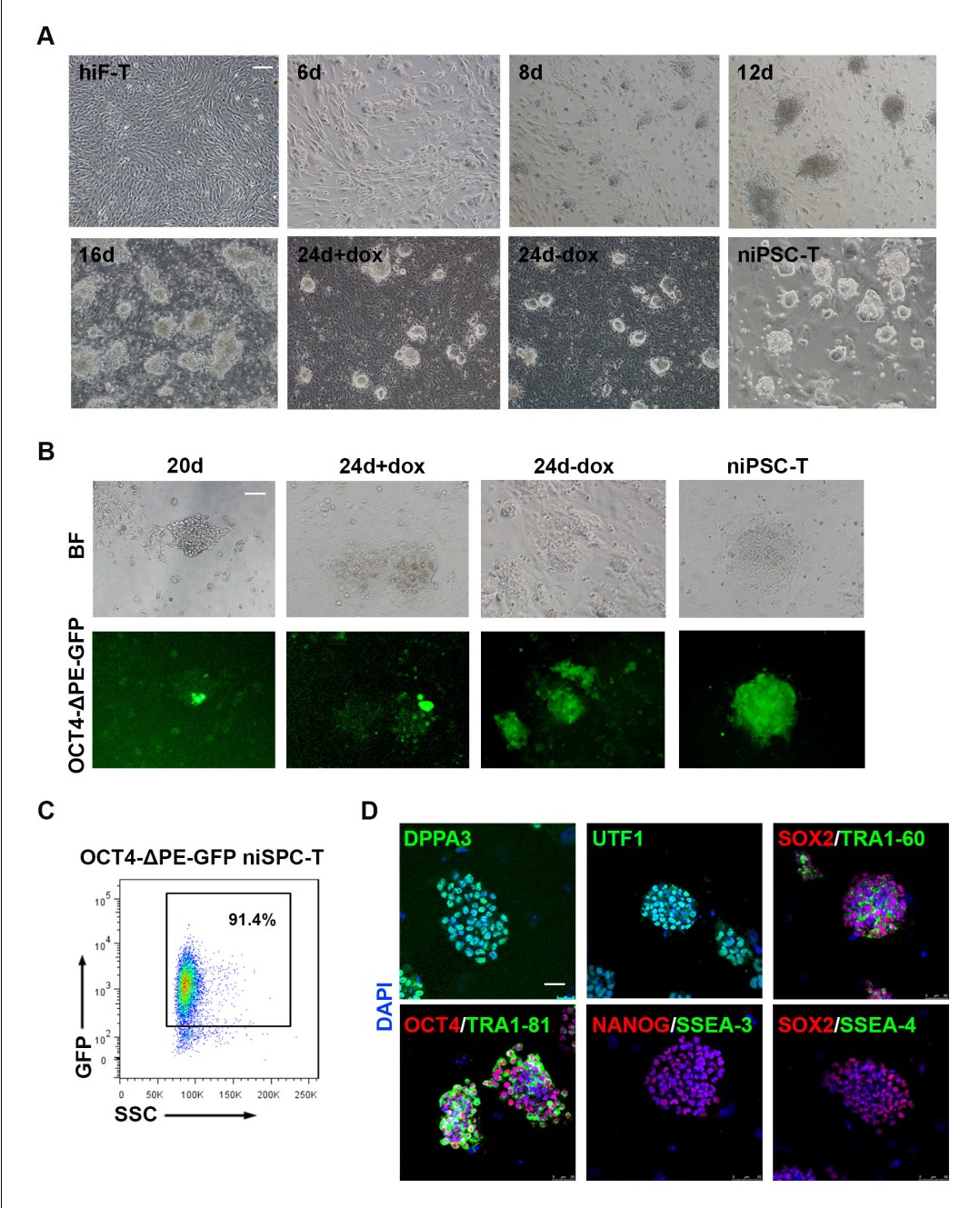

**Figure 1.** Establishment of the secondary naïve iPSC induction system.  (A) Representative bright field images of hiF-Ts, niPSC-Ts and reprogramming cells at the indicated time points during reprogramming. Scale bar, 100 μm. (B) Phase and OCT4-ΔPE-GFP images of niPSC-Ts and reprogramming cells at the indicated time points during reprogramming. Scale bar, 100 μm. (C) Flow cytometry analysis of the proportion of GFP+ cells in OCT4-ΔPE-GFP niPSC-Ts. (D) Immunostaining images of pluripotency-related marker expression in niPSC-Ts. Scale bar, 50 μm.

DOI: https://doi.org/10.7554/eLife.29518.002

The following figure supplement is available for figure 1:

**Figure supplement 1.** Optimization of secondary human naïve iPSCs reprogramming system.

DOI: https://doi.org/10.7554/eLife.29518.003

## Transcriptional profiling of naïve reprogramming cells

Next, we collected the mRNAs of cells at different time points throughout the naïve reprogramming process and performed RNA-seq analysis (*Figure 2A*). The Pearson correlation distance analysis of mRNAs segregated the cell samples into three distinct categories including hiF-T/0d/2d/6d, 8d/12d and 14d/20d/24d+dox/24d-dox/niPSC-T (*Figure 2—figure supplement 1A*). On the basis of the dynamics of the differentially expressed (DE) genes during naïve reprogramming (*Figure 2—figure supplement 1B*), multi-dimensional scaling (MDS) analysis exhibited a continuous trajectory of transcriptional dynamics from hiF-Ts to established niPSC-Ts (*Figure 2B*). However, distinct from the results for the primed reprogramming system (*Cacchiarelli et al., 2015*), the cellular states at days 20–24 during naïve reprogramming were similar, and dox withdrawal did not result in dramatic transcriptional changes in the cells on day 24 (*Figure 2B*), thus further suggesting the intrinsic differences between naïve and primed pluripotency. Compared with those in the primed reprogramming system, the epiblast-specific markers representing naïve pluripotency were gradually up-regulated (one-tailed t-test p-value=5.99e-22), whereas the primed-specific genes were gradually down-regulated (one-tailed t-test p-value=1.67e-3) during the naïve pluripotency induction process (*Figure 2—figure supplement 1C*). We also observed transient up-regulation followed by marked down-regulation of transcriptional factors OTX2 and ZIC2 during naïve reprogramming, which were known to direct OCT4 to primed state-specific enhancer sites (*Buecker et al., 2014*) and exhibited robust activation during primed reprogramming (*Figure 2—figure supplement 1D*).

Next, we characterized the transcriptome dynamics during naïve reprogramming in detail. On the basis of the DE genes between two adjacent time points of RNA-seq data (*Figure 2—figure supplement 1B*), we identified several dynamic expression clusters and focused on seven major patterns in three categories (down-regulated, up-regulated and transiently up-regulated), which were defined according to gene ontology (GO) enrichment analysis, developmental cell identity, and expression pattern of marker genes (*Figure 2C*; *Figure 2—figure supplement 2*) (*Edgar et al., 2013*). The activation of OSKM down-regulated genes involved in cell junction and extracellular matrix (ECM) organization occurred in two waves, at day 12 and day 20 of naïve reprogramming (*Figure 2C*; *Figure 2—figure supplement 2*). More importantly, pluripotency-associated gene networks were also activated in two waves; the earlier one consisted of genes with an early embryogenesis signature such as gradual up-regulation of *CDH1* and *NANOG* from day 2, whereas the later one comprised genes with pre-implantation characteristics, such as *DPPA3* and *TFCP2L1* (*Figure 2C*; *Figure 2—figure supplement 2*). For the transiently up-regulated genes during naïve reprogramming, the first transient wave peaked at day eight and was enriched in genes characteristic of late embryogenesis and pattern specification, such as *LHX9* and the *HOX* cluster genes (*Figure 2C*; *Figure 2—figure supplement 2*). The second wave peaked around day 12–14 and included metabolism-associated genes, such as *IGF2* and *AFP* (*Figure 2C*; *Figure 2—figure supplement 2*). The third wave, which peaked during the late reprogramming process at approximately day 24, was enriched in genes important for gamete generation (Fisher's exact test p-value=1.934e-3) and sexual reproduction (Fisher's exact test p-value=3.517e-3), such as *OVOL1* and *CGB5* (*Figure 2C*; *Figure 2—figure supplement 2*). Notably, the gene expression program with pre-implantation-like signatures exhibited significantly different dynamics in naïve compared with primed reprogramming systems, which exhibited robust up-regulation along the naïve reprogramming process, peaking in niPSC-Ts (*Figure 2C, D*). However, such expression was lost upon dox withdrawal and iPSC-T derivation during primed reprogramming (*Figure 2D*) (*Cacchiarelli et al., 2015*), results consistent with the MDS analysis results in both reprogramming systems, respectively (*Figure 2B*) (*Cacchiarelli et al., 2015*). In addition, in comparing the transcriptional profiles between the naïve reprogramming process of hiF-Ts to niPSC-Ts and human early embryo development (*Yan et al., 2013*), we found that the reprogramming cells at day 20 and 24 and niPSC-Ts most closely resembled human embryos at the late blastocyst stage (*Figure 2E*). Immunostaining of the pre-implantation marker DPPA3 and early embryogenesis marker UTF1 in the reprogramming cells at day 20 and day 24 in both induction systems also confirmed our observations in the transcriptional profiles (*Figure 2F*). Therefore, in contrast to the primed reprogramming system, in which the transcriptional profile finally dropped to the post-implantation-like stage (*Cacchiarelli et al., 2015*), the derivation of naïve iPSCs was accompanied by ordered waves of gene network activation, with the gene program finally stabilizing at the pre-implantation-like stage.

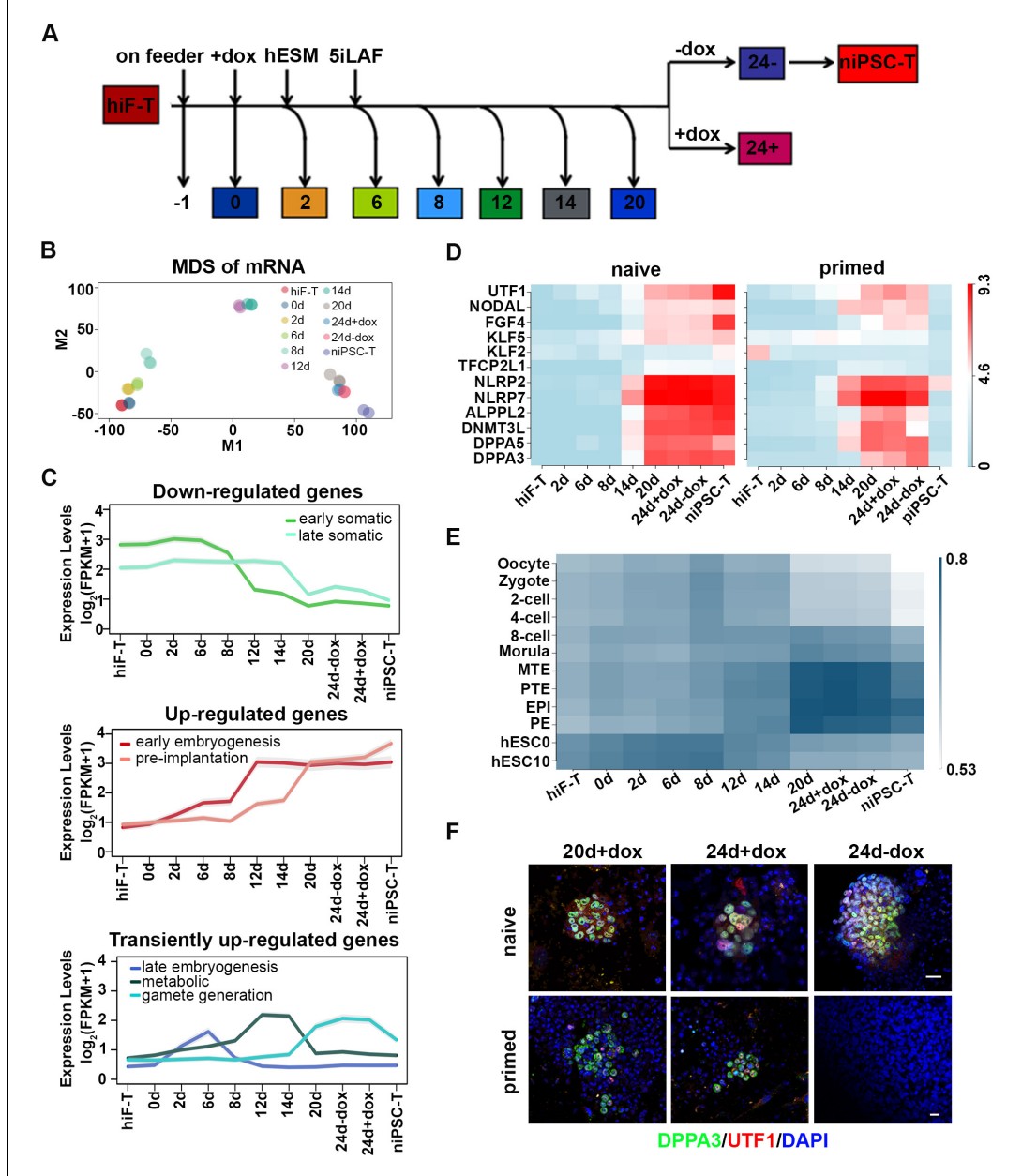

**Figure 2.** Transcriptional profiling of cells during naïve reprogramming. (A) Schematic representation of reprogramming intermediate collection at different time points, as indicated. hiF-T cells were first cultured in conventional hESM with dox for 6 days and then switched to 5iLAF culture medium supplemented with dox until day 20. Cells with or without dox treatment for four additional days were collected. (B) MDS analysis of RNA-seq data during the naïve reprogramming process. (C) Line plots showing transcriptional dynamics of differentially expressed genes during the naïve reprograming process. Genes were grouped by k-means clustering. Gray shades represent a 95% bootstrap confidence interval around the mean value. (D) Heatmaps showing the expression patterns of genes with pre-implantation signatures in both the naïve and primed reprogramming process. (E) Correlation analysis of transcriptional profiles between naïve reprogramming and the embryonic development process, with the Pearson correlation coefficient of each pair shown on each cell of the heatmap. (F) Immunostaining images of pluripotency-related marker expression in the reprogramming cells at indicated time points during naïve and primed reprogramming. Scale bar, 50 µm.

DOI: https://doi.org/10.7554/eLife.29518.004

The following figure supplements are available for figure 2:

**Figure supplement 1.** Transcriptional profiling of naïve pluripotency reprogramming cells.
DOI: https://doi.org/10.7554/eLife.29518.005

**Figure supplement 2.** Expression dynamics of gene clusters in *Figure 2C*.
DOI: https://doi.org/10.7554/eLife.29518.006

## Gene network activation with 8-cell-stage-like characteristics during naïve reprogramming

During human embryo development, the major wave of embryonic genome activation (EGA) occurs at approximately day three at ~8 cell (8C) stage (*Niakan et al., 2012*; *Vassena et al., 2011*), times corresponding to the major wave of zygotic genome activation (ZGA) at the 2 cell stage in mice, which is a key transcriptional feature of totipotency (*Latham and Schultz, 2001*; *Macfarlan et al., 2012*; *Schultz, 2002*). By analyzing single cell RNA-seq datasets of human early embryos (*Yan et al., 2013*), we identified 538 genes with expression levels peaking at the 8C stage during human embryonic development (*Figure 3A*), including the previously reported EGA regulators with PRD-like homeodomains such as *DUXA*, *OTX2* and *LEUTX* (*Töhönen et al., 2015*). Further investigation revealed that these 8C-genes were significantly enriched in the cluster including genes important for gamete generation (Fisher's exact test p-value=3.784322e-08) (*Figure 3A*; *Figure 3—figure supplement 1A*, *Figure 2—figure supplement 2*) and exhibited similar transiently up-regulated expression patterns during the late stages of naïve reprogramming (*Figure 3B*). In addition to *ZSCAN4* (*Figure 3C*; *Figure 3—figure supplement 1B*), whose mouse homologs are 2C-stage restricted and are important for telomere stability in mouse ESCs and iPSCs (*Ko, 2016*; *Zalzman et al., 2010*); we also identified *KLF17*, *TBX20* and some *PRAMEF* family genes including *PRAMEF15*, *PRAMEF5 etc.* in the 8C-stage category with transiently enhanced transcription activity during late stages of naïve reprogramming but not in the primed reprogramming system (*Figure 3—figure supplement 1C*). More importantly, we identified *MBD3L2/3/4/5* genes as 8C-genes during human embryo development (*Figure 3—figure supplement 1B*); these genes are the homologs of the mouse *Mbd3l2* gene, which is specifically expressed at the 2C stage during mouse development (*Jiang et al., 2002*). Similar to the genes listed above, the transcriptional dynamics of *MBD3L2/3/4/5* also showed dramatically increased expression during the late stages of naïve reprogramming from day 20 to 24 even after dox withdrawal, then decreased sharply after niPSC-T derivation (*Figure 3C, D*; *Figure 3—figure supplement 1C, D*). However, during the primed reprogramming, we observed only a small transient wave of *ZSCAN4* expression during the late stages from day 20 to day 24 (*Figure 3C*), and only marginal *MBD3L2/3/4/5* expression throughout the induction process (*Figure 3D*). Interestingly, the *MBD3L2/3/4/5* genes are loci-clustered in the human genome and are far from *MBD3L1*. Although these genes exhibited differential mRNA expression levels, the amino acid sequences of each gene in the cluster are identical, thus suggesting that the gene cluster might have evolved from mouse *Mbd3l2* via the copy-paste mode, similar to the situation observed in mouse *Zscan4* retrotransposons and ERV repeats. Using specific human ZSCAN4 and MBD3L2-5 antibodies, we performed immunostaining in cells during reprogramming. While only weak expression of ZSCAN4 as well as no expression of MBD3L2-5 could be detected around day 14, we could observe robust expression of ZSCAN4 and MBD3L2-5 around day 24 during naïve reprogramming (*Figure 3E*). During primed reprogramming, we could observe weak expression of ZSCAN4 and DPPA3 at 24d + dox, which diminished after dox withdrawal (*Figure 3—figure supplement 1E*). However, the expression of MBD3L2-5 could not be detected during primed reprogramming process (*Figure 3—figure supplement 1E*). Western blot analysis of MBD3L2-5 also confirmed our observations in both immunostaining and transcriptional profiles (*Figure 3F*). Although the transcriptional profiles of MBD3L cluster genes exhibited sharply decreased mRNA expression in the derived niPSC-T lines, we could still observe MBD3L2-5[+] cells in naïve iPSC clones compared to their absence in primed iPSCs by immunostaining (*Figure 3—figure supplement 1F*), suggesting the expression of MBD3L2/3/4/5 as an indicator for naïve pluripotency. In summary, the transient expression of 8C-genes, which occurs specifically during the late stages of naïve reprogramming, suggests the emergence of gene network activation with 8C-stage-like characteristics. Compared with the primed reprogramming that finally stabilized at the post-implantation-like state (*Cacchiarelli et al., 2015*), naïve reprogramming not only reached the pre-implantation-like status but more importantly, underwent reactivation of gene programs with human EGA-like characteristics.

Several lines of evidence in mouse pre-implantation development revealed activation of many retrotransposons, including ERVs, LINE-1 elements and SINE elements during ZGA at the 2C stage (*Peaston et al., 2004*). According to our observation of the 8C-gene network wave during naïve reprogramming, we next assessed the dynamics of transposon elements (TEs) during this process. We identified TEs that were specifically and highly expressed at the 8C stage (8C-TEs) during human

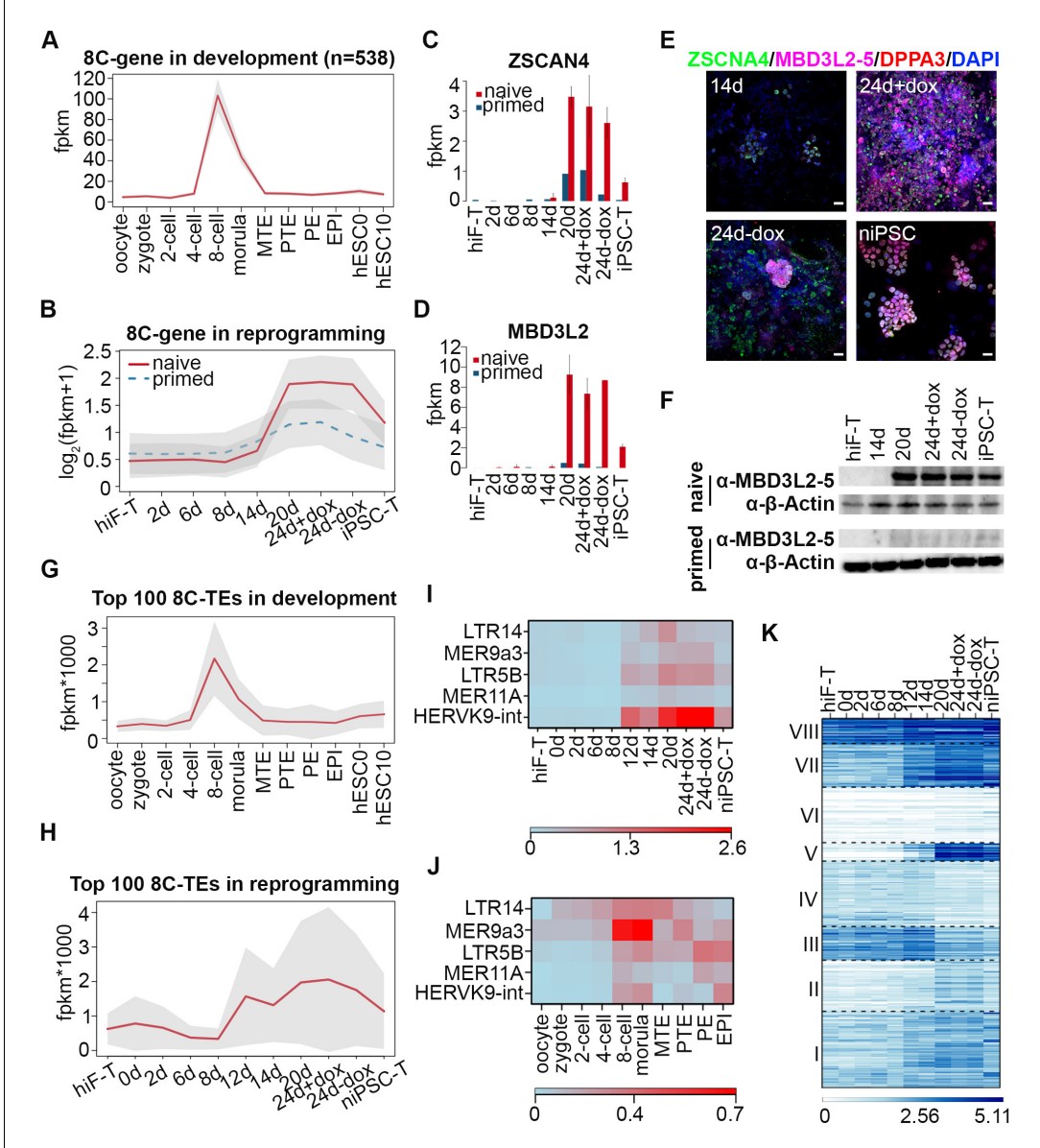

**Figure 3.** Transient activation of transcripts with 8C-stage-like signatures during naïve reprogramming. (A) Line plot showing expression dynamics of 8C-stage-specific genes during human embryonic development. Gray shades represent a 95% confidence interval around the mean value. (B) Line plot showing transcriptional dynamics of 8C-specific genes across naïve and primed reprogramming. (C–D) Bar plot showing the absolute expression values of *ZSCAN4* (C) and *MBD3L2* genes (D) in the naïve and primed reprogramming processes. Error bars represent a 95% confidence interval around the mean value. (E) Immunostaining images of ZSCAN4, MBD3L2-5 and DPPA3 expression in cells during naïve reprogramming. Scale bar, 50 μm. (F) Western blot results of MBD3L2-5 expression in naïve and primed reprogramming cells, niPSC-Ts and piPSC-Ts. β-ACTIN was used as endogenous control. (G) Line plot showing the expression patterns of 8C-specific TEs during human embryonic development. Gray shades represent a 95% confidence interval around the mean value. (H) Line plot showing expression dynamics of 8C-specific TEs during naïve reprogramming. (I–J) Heatmap of expression patterns of 8C-specific HERVK integrants across naïve reprogramming (I) and pre-implantation development (J). (K) Heatmaps showing different expression patterns of *KRAB-ZNF* genes in the naïve reprogramming process. K-means clustering was performed on *KRAB-ZNF* genes with k = 8 using R library 'amap'. Distance between genes was measured based on their correlation.

DOI: https://doi.org/10.7554/eLife.29518.007

The following figure supplements are available for figure 3:

**Figure supplement 1.** Dynamics of 8C-genes during naïve reprogramming.
DOI: https://doi.org/10.7554/eLife.29518.008

**Figure supplement 2.** Dynamics of TEs during naïve reprogramming.
DOI: https://doi.org/10.7554/eLife.29518.009

embryo development (*Figure 3G*); these TEs also showed transient reactivation during the late stages of naïve reprogramming (*Figure 3H*). Further investigation of these 8C-TEs showed a significant enrichment (Fisher's exact test p-value=0.00408) in LTR (*Figure 3—figure supplement 2A*). Notably, integrants of the HERVK-family that were activated upon EGA during development (*Grow et al., 2015*), especially MER9a3-HERVK9, were significantly up-regulated at day 12 of naïve reprogramming and down-regulated upon niPSC-T derivation (*Figure 3I*; J). These results, together with the recent observations of the transient activation of 2C-specific MERVL/*ZSCAN4* transcriptional network in the intermediate-late stages of mouse iPSC reprogramming (*Eckersley-Maslin et al., 2016*), suggest that the transient wave of 8C-transcripts is an important feature of human naïve reprogramming and that the reactivation of a gene network with the human EGA-like signature occurs during this process.

Recent studies in naïve pluripotency evaluation have indicated that naïve human ESCs display a unique transposon signature of cleavage-stage embryos with significant overexpression of the SINE-VNTR-*Alu* (SVA) family of transposon elements (*Theunissen et al., 2016*). In our system, reprogramming to naïve pluripotency induced significant up-regulation of several subgroups of the SVA family highly expressed during the morula-stage of human early embryo development, with the highest expression levels in niPSC-Ts (*Figure 3—figure supplement 2B*). Similar to previous observations (*Theunissen et al., 2016*), LTR7 and HERVH-int in the LTR7-HERVH family that were highly expressed in hESC0 and hESC10 that simulated the post-implantation stages, with no enrichment in 8C or morula stages during development (*Figure 3—figure supplement 2C*), exhibited higher expression levels across primed reprogramming than across naïve induction (*Figure 3—figure supplement 2D*). Hence, the transcriptional dynamics of TEs also revealed an ordered reactivation of transcripts with 8C- and morula-stage signatures in the intermediate-late stages of naïve reprogramming.

We also assessed the transcriptional profiles of KRAB-ZNF genes during reprogramming, which have been reported to play central roles in repressing TEs during early embryogenesis (*Huntley et al., 2006*; *Quenneville et al., 2011*). Transcriptional dynamics analyses divided the *KRAB-ZNFs* in to eight distinct clusters, among which the expression pattern of genes in cluster v was remarkably up-regulated in the naïve reprogramming system with robust expression in niPSC-Ts (*Figure 3K*). These genes included *ZNF534*, the repressor of LTR7-HERVH (*Figure 3—figure supplement 2E*). These results suggested that the naïve-specific KRAB-ZNF genes were activated to repress the TE network with a post-implantation-like signature, accompanied by the reactivation of TEs with characteristics reminiscent of morula-stage embryos during reprogramming to the naïve pluripotent state.

## Dynamic changes in epigenetic modifications in the naïve reprogramming system

Transcriptional profiling across the naïve reprogramming process revealed an ordered reactivation of diverse developmental pathways from late embryogenesis to the pre-implantation stages. Next we examined the dynamics of epigenetic modifications during this process. In contrast to the continuous hypermethylation during primed reprogramming, a significant decrease in global DNA methylation was observed throughout the naïve pluripotency induction process (*Figure 4A*; *Figure 4—figure supplement 1A*), including an average methylation level resembling that in ICM by the end of reprogramming in niPSC-Ts, as previously demonstrated (*Figure 4A*) (*Leitch et al., 2013*; *Okae et al., 2014*; *Pastor et al., 2016*; *Smith et al., 2014*; *Theunissen et al., 2016*). Quantitative analysis via HPLC-MS also revealed a dramatic decrease in 5mC levels throughout naïve reprogramming but not the primed reprogramming process (*Figure 4—figure supplement 1B*). Dynamic changes in differentially methylated C sites (DMCs) throughout naïve reprogramming also showed an increasing trend of hypomethylated C site ratios (*Figure 4—figure supplement 1C*), a result consistent with the global demethylation trend (*Figure 4A*; *Figure 4—figure supplement 1A, B*). For detailed analysis, we characterized naïve-specific differentially methylated Regions (DMRs) by comparing the DMRs of niPSC-Ts with those of hiF-Ts and piPSC-Ts (*Figure 4—figure supplement 1D*); the results showed a dramatic trend of down-regulation in the average methylation ratios during naïve reprogramming, but no significant changes during primed reprogramming (*Figure 4B*). We also found that the identified naïve-specific DMRs were enriched in genes related to naïve

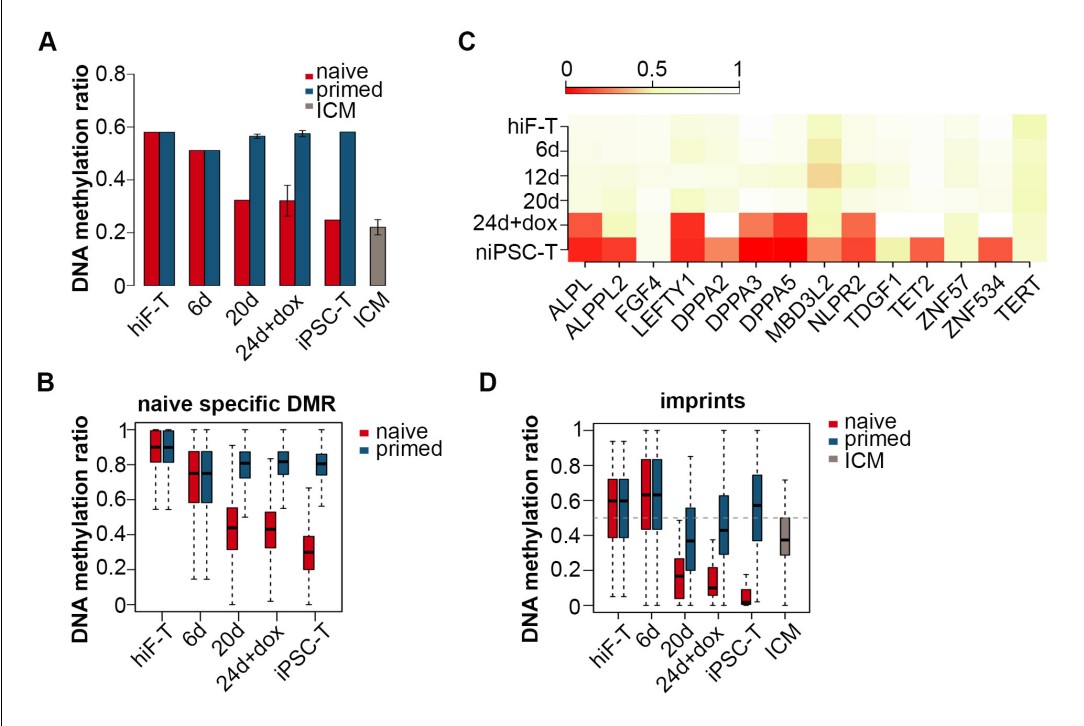

**Figure 4.** Changes in DNA methylation during naïve reprogramming. (**A**) Bar plot showing changes in average DNA methylation ratios of all covered C sites during naïve and primed reprogramming. Error bars represent a 95% confidence interval around the mean value. (**B**) Box plot showing DNA methylation ratio dynamics of naïve specific DMRs during naïve and primed reprogramming. The middle lines of the boxes indicate the median, the outer edges represent the first and the third quartiles, and the whiskers indicate the 1.5 × interquartile range below the lower quartile and above the upper quartile. (**C**) Dynamics in the DNA methylation levels of naïve-specific DMR-related genes during naïve reprogramming. (**D**) DNA methylation over stable primary imprints during naïve and primed reprogramming. The middle lines of the boxes indicate the median, the outer edges represent the first and the third quartiles, and the whiskers indicate the 1.5 × interquartile range below the lower quartile and above the upper quartile.
DOI: https://doi.org/10.7554/eLife.29518.010

The following figure supplement is available for figure 4:

**Figure supplement 1.** Dynamics of DNA methylation in naïve reprogramming.
DOI: https://doi.org/10.7554/eLife.29518.011

pluripotency, which exhibited significant DNA de-methylation at the promoter regions on approximately day 24 during reprogramming (*Figure 4C*).

Growing evidence indicates that naïve PSCs lose DNA methylation at primary imprints, which are retained throughout pre-implantation development (*Pastor et al., 2016*). We examined DNA methylation dynamics in 31 stable primary imprints in both the naïve and primed reprogramming processes. The DNA methylation ratios markedly decreased during naïve reprogramming, and methylation was nearly completely lost in niPSC-Ts (*Figure 4D*), results similar to previous observations (*Pastor et al., 2016*). However, in primed reprogramming, the decrease was more moderate, and the DNA methylation ratio finally stabilized at ~50% in piPSC-Ts, as previously reported (*Figure 4D*) (*Pastor et al., 2016*). The failure to reactivate DNMT3A and DNMT3B, as well as the over-activation of TET2 (*Figure 4—figure supplement 1E*), might be causative of the genome-wide hypomethylation and loss of imprinting observed in the naïve pluripotency induction process.

To assess the genome-wide landscape of histone modifications, we mapped two active modifications (H3K4me2 and H3K4me3) and two repressive modifications (H3K27me3 and H3K9me3) during naïve reprogramming. We first classified genes to 'H3K4me2-only', 'H3K4me3-only' and 'both H3K4me2/H3K4me3' catalogs based on the H3K4me3 and H3K4me2 modification signals detected on the promoter region of each gene. A transiently decreased pattern of 'both H3K4me2/H3K4me3' gene numbers was observed during this process, with an increasing pattern of 'H3K4me3-only' gene numbers as well as a decreasing pattern of 'H3K4me2-only' signals, thus suggesting that there was a

transient transition from H3K4me2 to H3K4me3 on the promoter regions during the naïve reprogramming process (*Figure 5—figure supplement 1A*).

Next, we focused on H3K4me3 and H3K27me3 dynamics during reprogramming. We observed increasing numbers of bivalent genes during the reprogramming process (*Figure 5A,B*), however, the up-regulation of bivalent signals was much slower with less gene numbers in niPSC-Ts during naïve reprogramming compared to primed reprogramming (*Figure 5A,B*). For further comparison, we clustered the genes with similar patterns of bivalency on their promoters in both naïve and primed reprogramming process, and found that there were two clusters of genes highly related to embryonic development showing increasing bivalent signals on their promoter regions during primed reprogramming, while exhibiting almost no H3K4me3/H3K27me3 signals during the whole reprogramming process to naïve pluripotency (*Figure 5C*; *Figure 5—figure supplement 1B*), results consistent with the previous report (*Pastor et al., 2016*; *Theunissen et al., 2014*; *Yang et al., 2016*).

To address the effect of histone modification on transcriptional activity, we also examined the H3K4me3 and H3K27me3 modification dynamics around the transcription start sites (TSSs) of gene clusters with different expression patterns. The average H3K4me3 signals were highly correlated with the transcriptional dynamics and increased around the TSSs of genes with pre-implantation and early embryogenesis signatures, while the average H3K27me3 signals decreased around the TSSs of these genes during reprogramming (*Figure 5D*).

We also checked the dynamics of H3K9me3 modification and found a decreasing pattern of H3K9me3 signals during naïve reprogramming, including integrants of SVA family, the naïve specific TEs (*Figure 5E*; *Figure 5—figure supplement 1C*). Moreover, the H3K9me3 signals around the 8C-TEs, especially MER9a3-HERVK9, exhibited a transiently decreased pattern at day 14, strongly correlated with the transiently increasing transcriptional activity of these 8C-TEs during naïve reprogramming (*Figure 5F*).

## Integrative analysis of the naïve iPSC reprogramming system

Next, we focused on the down- and up-regulated patterns of genes with early/late somatic, early embryogenesis- and pre-implantation-like signatures (*Figure 2C*; *Figure 2—figure supplement 2*) and analyzed the relationships among the gene expression patterns, DNA methylation status and histone modifications. Despite their distinct expression dynamics, the transcriptional levels of these genes were closely correlated with the changes in epigenetic modifications at the promoter regions during naïve reprogramming, regulated by DNA methylation, histone modifications, or both (*Figure 6A*). Deep investigations at the base level also confirmed our observations above (*Figure 6—figure supplement 1A*). For detailed analysis, we divided the up-regulated genes during reprogramming into three groups: high-CpG-density promoters (HCPs), intermediate-CpG-density promoters (ICPs) and low-CpG-density promoters (LCPs) on the basis of the CpG ratios and the GC contents of their promoters (*Figure 6—figure supplement 1B*). We observed that, the two clusters with up-regulation trend showed different CpG-density patterns on these promoters (*Figure 6A*). Furthermore, compared with the genes with lower CpG-densities in their promoters, genes with higher CpG-densities tended to undergo more rapid DNA demethylation and re-establishment of active histone modifications (H3K4me2/3), as well as earlier up-regulation in transcription during naïve reprogramming (*Figure 6B*; *Figure 6—figure supplement 1B*). We examined the genes classified by different CpG-densities at promoter regions and found that the genes associated with core naïve pluripotency were enriched primarily in the LCP and ICP groups (*Figure 6—figure supplement 1C*). Together, these results indicated that the transcriptional dynamics strongly correlated with epigenetic changes during naïve reprogramming, which may be affected by the CpG density at gene promoter regions.

## Discussion

The newly discovered 'naive' state of human pluripotency holds great promise for early embryo development studies and therapeutic manipulations, overcoming the application bottlenecks of pluripotent stem cells at primed state. However, recent studies have primarily focused on naïve pluripotency derivation and identification (*Dodsworth et al., 2015*; *Gafni et al., 2013*; *Hanna et al., 2010*; *Huang et al., 2014*; *Takashima et al., 2014*; *Theunissen et al., 2016*; *Theunissen et al., 2014*), in lack of in-depth mechanical studies of naïve pluripotency establishment during reprogramming.

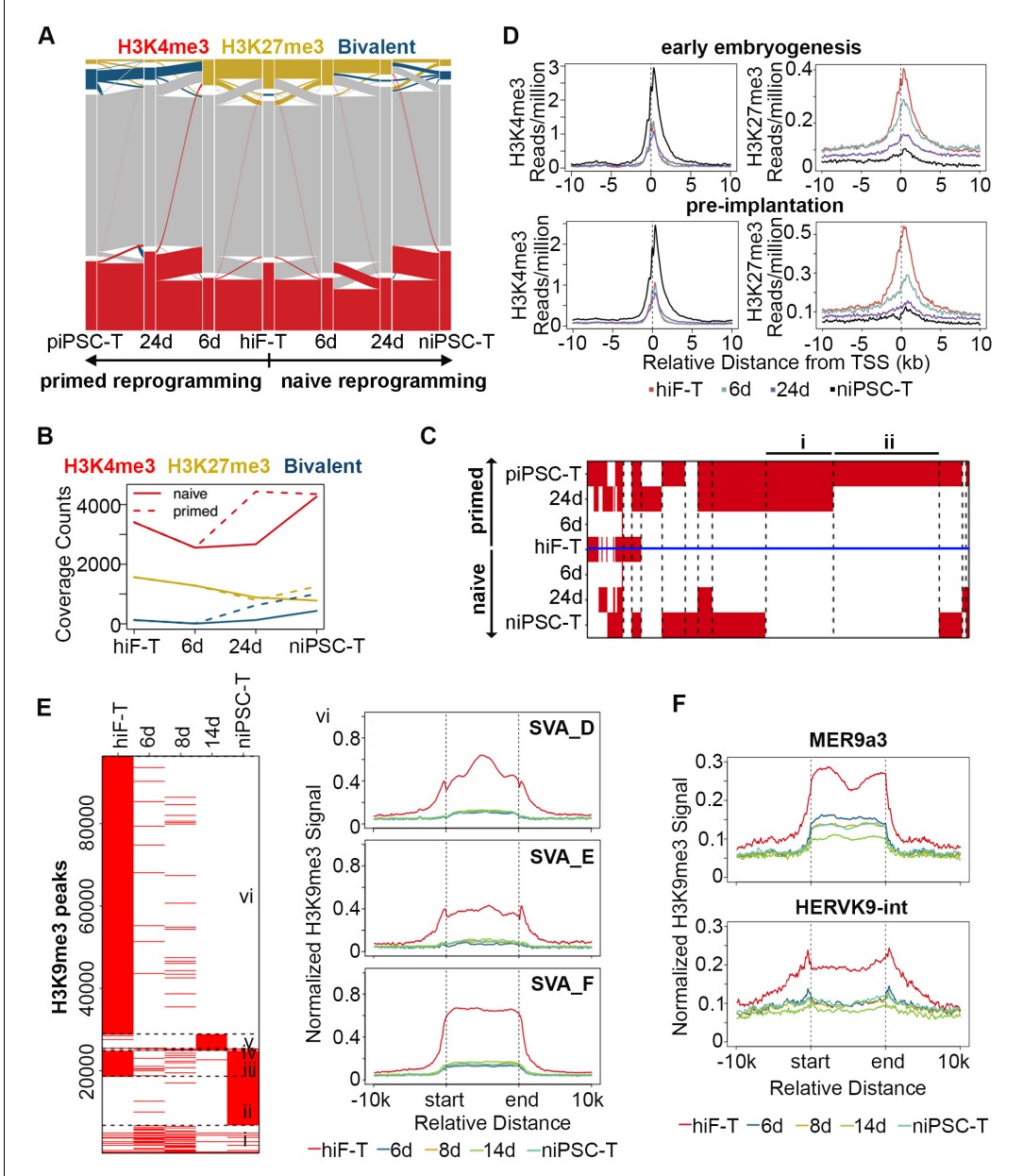

**Figure 5.** Histone modification profiles during naïve reprogramming. (A) Alluvial plots showing the global dynamics of genes covered by different chromatin states during naïve reprogramming. Each line represents a gene. Red bar represents genes with promoter that covered only by H3K4me3. Yellow bar represents genes with promoter that covered only by H3K27me3. Blue bar represents genes with promoter that covered by both H3K4me3 and H3K27me3. Grey bar represents genes with promoter that covered by neither H3K4me3 nor H3K27me3. (B) Line plot showing dynamics of different histone modification signals across naïve and primed reprogramming. (C) Heatmap showing clusters of genes with different bivalency patterns across naïve and primed reprogramming. (D) Average profiles of H3K4me3 and H3K27me3 signals surrounding the TSS of genes characteristic for early embryogenesis and pre-implantation during naïve reprogramming. (E) Heatmap showing six clusters of H3K9me3 peaks with different patterns (left panel) and average H3K9me3 profiles around integrants of SVA family in cluster vi of the heatmap (right panel) during naïve reprogramming. (F) Average H3K9me3 profiling around MER9a3-HERVK-9 TE during naïve reprogramming.

DOI: https://doi.org/10.7554/eLife.29518.012

The following figure supplement is available for figure 5:

**Figure supplement 1.** Dynamics of histone modifications in naïve reprogramming.

DOI: https://doi.org/10.7554/eLife.29518.013

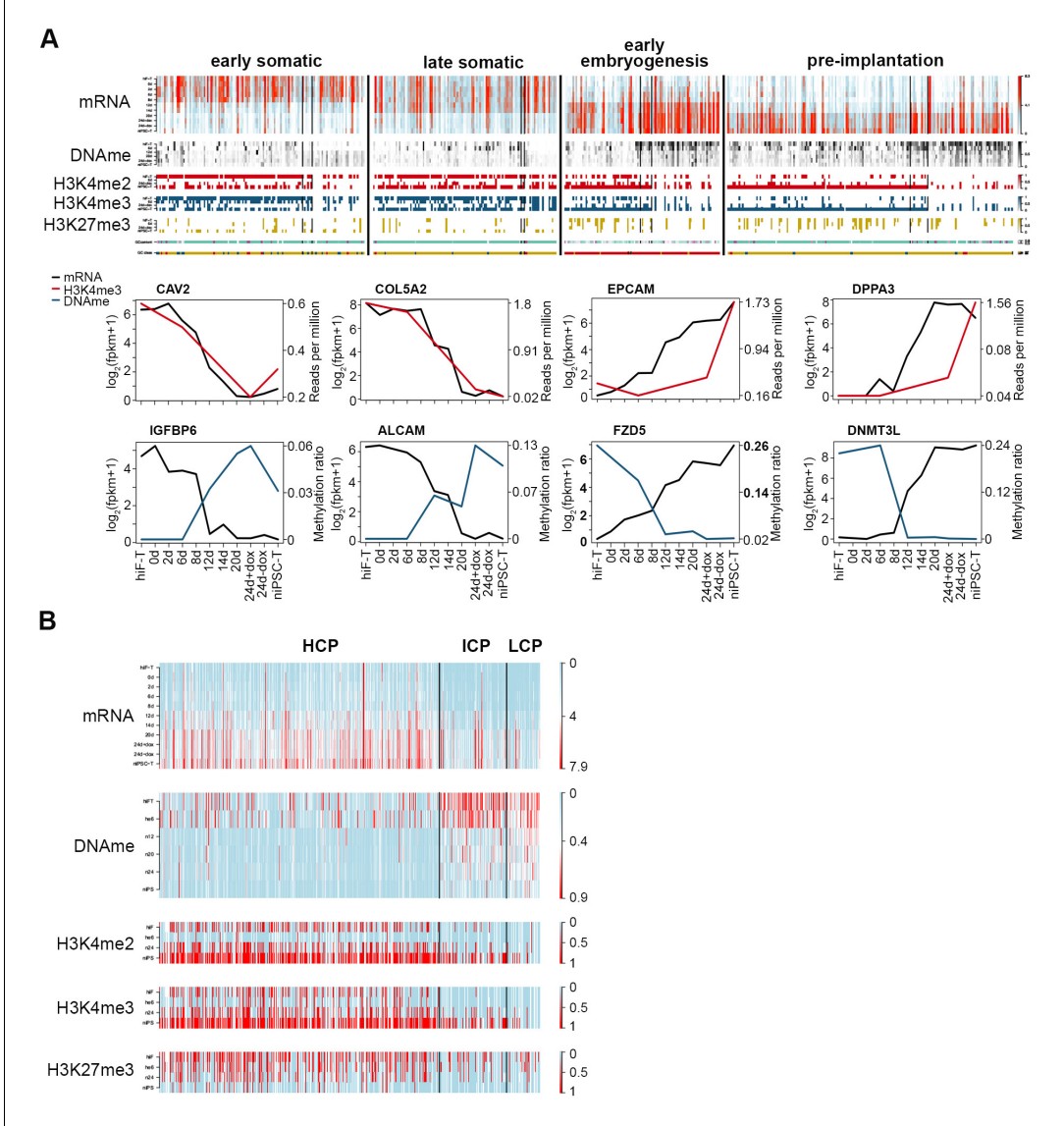

**Figure 6.** Integrative analysis of transcriptional and epigenetic dynamics during naïve reprogramming. (**A**) Transcriptional dynamics of genes with different patterns are closely correlated with epigenetic modifications at promoter regions during naïve reprogramming. Black lines in the heatmaps separate genes regulated by histone modifications (upper panel, left part), DNA methylation (upper panel, right part) or both (upper panel, middle part). Line plots (lower panels) show the representative genes in each category that are regulated by H3K4me3 modification or DNA methylation. (**B**) Up-regulated genes with different CG ratios in their promoters exhibit different kinetics with regard to transcription, DNA methylation and H3K4me2/H3K4me3/H3K27me3 coverage during naïve reprogramming.

DOI: https://doi.org/10.7554/eLife.29518.014

The following figure supplement is available for figure 6:

**Figure supplement 1.** Epigenetic changes of representative genes during naïve reprogramming path.
DOI: https://doi.org/10.7554/eLife.29518.015

Here, by using the secondary human naïve reprogramming system, we monitored the dynamics of transcriptome and epigenome during naïve pluripotency induction process. We observed ordered reactivations of transcriptional networks sharing signatures reminiscent of reversed embryonic development, including a significant transient wave of 8C-stage-specific transcripts expression in the late reprogramming stages followed by the stabilization of the transcriptome with pre-implantation-like characteristics, thus suggesting that the activation of a network with human EGA-like characteristics occurs during naïve pluripotency induction.

Recent advances have demonstrated that during the primed iPSCs induction, the pre-implantation-like state is reached quickly but is lost upon dox withdrawal and iPSC line derivation (*Cacchiarelli et al., 2015*). Distinctly from that phenomenon, the reprogramming cells in the naïve pluripotency induction process undergo a gradual reactivation of networks with pre-implantation signatures (*Figure 2*). Interestingly, the expression wave of 8C transcripts observed specifically in naïve reprogramming (*Figure 3*) suggests that the 8C-stage-like state is transiently established during the naïve pluripotency induction process. We also observed a significant genome-wide DNA demethylation as early as 12 days after naïve induction, which eventually led to hypomethylation in naïve iPSCs with similar methylation ratios of ICM in vivo, as previously reported (*Hanna et al., 2010*; *Pastor et al., 2016*; *Theunissen et al., 2016*) (*Figure 4A*; *Figure 4—figure supplement 1A*). The failure to maintain DNA methylation might be correlated with the lower expression levels of de novo DNA methyltransferase *DNMT3A/3B*, as well as the higher expression level of 5mC oxidase *TET2*, across naïve reprogramming compared with the primed system (*Figure 4—figure supplement 1E*).

Taken together, dissecting and analyzing the dynamics of naïve induction process provide the first molecular roadmap of the reprogramming of human somatic cells into naïve pluripotent state, which improve the understating of the molecular networks in the establishment and maintenance of naïve pluripotency and provide a theoretical basis for further applications.

## Materials and methods

### Key resources table

| Reagent type (species) or resource | Designation | Source or reference | Identifiers | Additional information |
|---|---|---|---|---|
| gene () | NA | NA | | |
| strain, strain background () | NA | NA | | |
| genetic reagent () | NA | NA | | |
| cell line () | Human embryonic fibroblasts (HEFs); Primary primed iPSC lines; hiF-T cell lines; Secondary primed iPSC lines; Secondary naïve iPSC lines | This paper; Cacchiarelli, D., Trapnell, C., Ziller, M.J., Soumillon, M., Cesana, M., Karnik, R., Donaghey, J., Smith, Z.D., Ratanasirintrawoot, S., Zhang, X., et al. (2015). Cell. 2015 Jul 16;162(2):412–424. doi: 10.1016/j.cell.2015.06.016. Yan, L., Yang, M., Guo, H., Yang, L., Wu, J., Li, R., Liu, P., Lian, Y., Zheng, X., Yan, J., et al. (2013). Nat Struct Mol Biol. 2013 Sep;20(9):1131–9. doi: 10.1038/nsmb.2660. Epub 2013 Aug 11. | | |
| transfected construct () | dox-inducible, polycistronic OKMS lentiviral vector | Addgene 51543. Cacchiarelli, D., Trapnell, C., Ziller, M.J., Soumillon, M., Cesana, M., Karnik, R., Donaghey, J., Smith, Z.D., Ratanasirintrawoot, S., Zhang, X., et al. (2015).Cell. 2015 Jul 16;162(2):412—424. doi: 10.1016/j.cell.2015.06.016. | | |
| biological sample () | hiF-T/0d/2d/6d/8d/12d/14d/20d/24d+dox/24d-dox/niPSC-T; Oocyte/Zygote/2 cell/4 cell/8 cell/Morula/MTE/PTE/EPI/PE/hESC0/hESC10; hiF-T/2d/6d/8d/14d/20d/24d+dox/24d-dox/piPSC-T; | this paper; | | |

*Continued on next page*

*Continued*

| Reagent type (species) or resource | Designation | Source or reference | Identifiers | Additional information |
|---|---|---|---|---|
| antibody | anti-SSEA3, SSEA4, TRA-1–60, UTF1, DPPA3, ZSCNA4, MBD3L2 | Millipore MAB4304, Millipore MAB4360, Abcam ab24273, Santa Cruz sc-67249, Millipore AB4340, Abcam ab174802 | | |
| recombinant DNA reagent | NA | NA | | |
| sequence-based reagent | KAPA Stranded mRNA-Seq Kit; KAPA DNA Library Preparation Kits | KAPA KK8401; KAPA KK8234 | | |
| peptide, recombinant protein | human LIF recombinant protein; bFGF recombinant protein | Peprotech 300–05; Peprotech 450–33 | | |
| commercial assay or kit | bowtie; TopHat; Cufflinks; edgeR; MACS2; | KAPA KK8401; KAPA KK8234 | | |
| chemical compound, drug | Activin A; PD0325901; IM-12; SB590885; WH-4–023; Y-27632; | Peprotech 120–14; Stemgent 04–0012; Enzo BML-WN102; R and D systems 2650; A Chemtek 0104–002013; Stemgent 04–0012; | | |
| software, algorithm | bowtie; TopHat; Cufflinks; edgeR; MACS2; | PMID: 22388286; PMID: 19289445; PMID: 22383036; PMID: 24743990; PMID: 18798982; | RRID:SCR_005476; RRID:SCR_013035; RRID:SCR_014597; RRID:SCR_012802; RRID:SCR_013291 | NA |
| other | | | | |

## Human skin tissue acquisition, cell culture and reprogramming

Human skin specimens from abortive fetus were obtained from the Clinical and Translational Research Center of Shanghai First Maternity and Infant Hospital, Tongji University to make human embryonic fibroblasts (HEFs). The identity of HEFs has been authenticated by STR profiling and the cells are cultured with no mycoplasma contamination.

HEFs were cultured in DMEM (Invitrogen) supplemented with 10% FBS (Invitrogen). Primed iPSCs were cultured in hESM containing DMEM/F12 with 20% knockout serum replacement (KSR) (Invitrogen) and 4 ng/ml bFGF (Peprotech). Naïve iPSCs were cultured in 5iLAF medium containing DMEM/F12: Neurobasal (1:1) (Invitrogen), 1% N2 supplement (Invitrogen), 2% B27 supplement (Invitrogen), 0.5% KSR (Invitrogen), 20 ng/ml human LIF (Peprotech), 8 ng/ml bFGF (Peprotech), 50 µg/ml BSA (Sigma) and the following cytokines and small molecules: PD0325901 (Stemgent, 1 µM), IM-12 (Enzo, 1 µM), SB590885 (R and D systems, 0.5 µM), WH-4–023 (A Chemtek, 1 µM), Y-27632 (Stemgent, 10 µM), and Activin A (Peprotech, 20 ng/ml), and passaged by Accutase (Sigma) every 4–5 days as previously reported (*Theunissen et al., 2014*; *Yang et al., 2016*).

For reprogramming, hEFs were infected with the dox-inducible, polycistronic OKMS lentiviral vector (addgene) and cultured in conventional human embryonic stem cell medium(hESM) to generate primary primed iPSCs, which were further differentiated into hiFs and then infected with pBabe-TERT retroviral vector and screened by 1.6 ng/ml puromycin to generate hiF-Ts. Secondary naïve iPSC-Ts were derived by culturing hiF-Ts in hESM with dox for 6 days followed by changing to 5iLAF medium with dox for 14 days, which were then maintained in 5iLAF medium without dox for additional 4 days. After 24 days of infection, mESC-like colonies were picked and expanded wihout dox on irradiated feeder cells in 5iLAF medium.

For differentiation from naïve to primed state, niPSC-T cells were first digested into single cells and plated onto irradiated feeder cells, which were then cultured in conventional hESM supplemented with Y-27632 (Stemgent, 10 µM) for 8–10 days.

## Detection of cells growth rate

$5 \times 10^4$ cells (early hiF, late hiF, early hiF-T and late hiF-T) were passaged onto 12-well plate in three replicates. Calculation of cell growth rate was performed by counting of cell numbers at 24 hr, 48 hr, 72 hr and 96 hr respectively.

## Antibodies

For immunostaining, the primary antibodies used included those against OCT3/4 (1:500, Santa Cruz), SOX2 (1:500, Santa Cruz), NANOG (1:500, Abcam), SSEA3 (1:50, Millipore), SSEA4 (1:50, Millipore), TRA-1–60 (1:50, Millipore), UTF1 (1: 200, Abcam), DPPA3 (1:50, Santa Cruz), ZSCAN4 (1:100, Millipore) and MBD3L2 (1:100, Abcam). The following secondary antibodies were used: Alexa Fluor 594-conjugated donkey anti-mouse IgG (1:500; Invitrogen), fluorescein isothiocyanate (FITC) 488-conjugated donkey anti-rabbit IgG (1:500; Invitrogen), and FITC 488-conjugated donkey anti-goat IgG (1:500; Invitrogen). Anti-H3K4me2 (Millipore), Anti-H3K4me3 (Millipore), Anti-H3K27me3 (Abcam) and Anti-H3K9me3 (Millipore) antibodies were used for ChIP experiments.

## Flow cytometry and immunofluroresence staining

For flow cytometry, OCT4-ΔPE-niPSC-Ts were collected, washed and re-suspended in FACS buffer containing PBS (Invitrogen) supplemented with 2% FBS (Invitrogen). All analyses were performed on a MoFloXDP cell sorter (Beckman Coulter). Flow cytometry data were processed using Flow Jo software.

Immunostaining was performed according to standard protocols. In brief, cells were fixed with PBS containing 4% paraformaldehyde (Sigma-Aldrich) overnight at 4°C and permeabilized for 15 min in PBS containing 0.5% Triton X-100. After incubation with blocking buffer (PBS containing 4% BSA) for 30 min at room temperature, cells were incubated with primary antibodies followed by secondary antibodies. Nuclei were stained with 4',6-diamidino-2-phenylindole (1:10,000; Sigma-Aldrich). Images were taken using A1 Nikon confocal microscope.

## RNA-seq library generation and sequencing

Total RNAs were isolated from naïve iPSCs and reprogramming cells using TRizol (Invitrogen). To generate RNA sequencing libraries, KAPA Stranded mRNA-Seq Kit (KAPA) was used following the manufacturer's instructions. Adapters were offered by TruSeq Library Prep Pooling kit (Illumina). Single-end 50 bp sequencing was further performed on a HiSeq 2500 or 2000 (Illumina) at Berry Genomics Corporation.

## RNA-seq data processing

Gene expression Analysis

All RNA-seq reads were aligned to the human genome (hg19) using TopHat (v2.0.12) with default parameters (*Trapnell et al., 2009*). Gene expression level was measured as FPKM using Cufflinks (v2.2.1) to eliminate the effects of sequencing depth and transcript length (*Trapnell et al., 2010*). MDS clustering analysis was based on expression profile of all genes using R function 'cmdscale'. For each comparison, differential expressed (DE) genes were founded using GFOLD(v1.1.3) with the GFOLD value >0.58 (fold change >1.5)(*Feng et al., 2012*). For following analysis, FPKM were log2 transformed after adding a pseudo-count of 1. K-means clustering was performed on combined DE genes of each nearby time points with *k = 14* using R library 'amap'. Distance between genes was measured based on their correlation. Batch effects of samples from different systems are removed using *removeBatchEffect* function of the R library 'edgeR'. The negative values in the normalized data are considered as zero. Pearson correlation coefficient was calculated between each two samples on common genes using R function cor().

## Retrotransposon expression analysis

All RNA-seq reads were aligned to the human genome (hg19) using bowtie2(v 2.2.3) with default parameters. Then FPKM of repeat classes were calculated as the sum of the number of reads that align to each class divided by the genome coverage of the class in kilobases. FPKM of each repeat elements were calculated same as repeat classes. Repeats that expressed in specific stages were

identified by comparing the repeat expression in the specific stage with the average expression. The repeats with average expression lower than 0.0001 were discarded.

A time-point specific expressed score was calculated as the expression of given time point divided by the average expression for each candidate gene, which should have a max expressed value in the given stage. The top 100 8C-TEs are the top 100 TE ranked by the time-point specific expressed score.

## Gene ontology analysis

Functional annotation was performed using the Database for Annotation, Visualization and Integrated Discovery (DAVID) Bioinformatics Resource (*Huang et al., 2009*). Gene ontology terms for each function cluster were summarized to a representative term and P-values were plotted to show the significance. The genes are classified according to expression pattern of marker genes and developmental cell identity using LifeMap Discovery (*Edgar et al., 2013*).

## ChIP-seq library generation and sequencing

Reprogramming cells and iPSCs were cross-linked and lysed to release chromatin, which were then sonicated by SonicsVibraCell Sonicator (Covaris) and immunoprecipitated with pretreated antibody-coupled ProteinG Dynabeads (Invitrogen) at 4°C for 8–12 hr. The ChIPed DNA was reverse-cross-linked, eluted, purified and quantified by Qubit dsDNA HS assay kit (Life Technologies). For ChIP sequencing, ChIP-seq libraries were prepared according to the protocols described in the KAPA DNA Library Preparation Kits (KAPA). Paired-end 125 bp sequencing was further performed on a HiSeq 2500 or 2000 (Illumina) at Berry Genomics Corporation.

## ChIP-seq data processing

All ChIP-seq reads were aligned to the human genome (hg19) using bowtie2(v 2.2.3) with default parameters. Then reads signal for each sample were generated using the MACS2 (v2.1.0.20140616) pileup function and were normalized to 1 million reads for visualization. Chromatin states were identified and characterized using ChromHMM (v1.11) (*Ernst and Kellis, 2012*). The reads alignment files of H3K4me2, H3K4me3 and H3K27me3 modifications across 4-time stages during reprogramming path were binned into 200 bp bins using the Binarize Bam command. Next, a model was trained using the Learn Model command with 200 bp resolution and default parameters. Finally, for active marker, the whole genome was classified into four states: H3K4me2-only (without H3K4me3) region, H3K4me3-only (without H3K4me2) region, H3K4me2 and H3K4me3 region and non-marked region at each stage. For bivalency analysis, the whole genome was then classified into four states: H3K4me3-only (without H3K27me3) region, H3K27me3-only (without H3K4me3) region, H3K4me3 and H3K27me3 region and non-marked region at each stage. Genes were classified based on their promoters overlapping with ChromHMM segments in each stage. Promoters were defined as ±2 kb around the TSS. Genes overlapped with multiply segments in one stage were discarded in following analysis.

Alluvial diagrams of reprogramming lineages were plotted using the alluvial function in R to show the transitions of different histone mark covered gene number. The gene number was the counter of genes in each state and the percentages of the specified intervals in each stage were plotted to show the global trend of that specific chromatin state. The alluvial diagrams showed the percentage changes of different chromatin states covered gene during each transition; the lines from the present stage to next stage cannot be traced, as they represent different genes.

## RRBS and data processing

For Reduced Representation Bisulfite Sequencing (RRBS), genomic DNAs were extracted from reprogramming cells and iPSCs using DNeasy Blood and Tissue Kits (QIAGEN), digested by (NEB) and inactivated by heating. The sequencing libraries were constructed as previously described (*Gu et al., 2011*). Paired-end 125 bp sequencing was further performed on a HiSeq 2500 or 2000 (Illumina) at Berry Genomics Corporation. All Reduced Representation Bisulfite Sequencing (RRBS) reads were aligned to human genome (hg19).

The correlation between DNA methylation ratio and gene expression was calculated using function cor() in R on each gene.

## CpG ratio calculation and promoter classification

Local CpG ratio was calculated for 500 bp bins with 50 bp steps, as previously defined (*Weber et al., 2007*). The CpG ratio for each transcript was calculated as the max local CpG ratio around ±2 kb of the TSS. The transcripts were then separated into high-CpG-density promoters (HCPs), intermediate-CpG-density promoters (ICPs) and low-CpG-density promoters (LCPs) based on the CpG ratio and the GC content cut-off previously defined (*Weber et al., 2007*).

## Accession number

The accession number for all the sequencing-derived data in this paper is GEO: GSE89072.

The accession number for RNA-seq data of embryo development used in this paper is GEO: GSE36552.

The accession number for RNA-seq data of primed cell reprogramming used in this paper is GEO: GSE62777.

The accession number for DNA methylation data of human early embryos used in this paper is GEO: GSE49828

# Acknowledgements

We thank our colleagues in the laboratory for their assistance with the experiments and comments on the manuscript. This work was supported by the Ministry of Science and Technology of China (2016YFA0100400, and 2014CB964601), the National Natural Science Foundation of China (NSFC) (31671530, 31325019 and 31471392), and the Science and Technology Commission of Shanghai Municipality (Grant 14YF1403900, 15XD1503500).

# Additional information

### Funding

| Funder | Grant reference number | Author |
| --- | --- | --- |
| National Natural Science Foundation of China | 31671530 | Yixuan Wang |
| National Natural Science Foundation of China | 31471392 | Yixuan Wang |
| Ministry of Science and Technology of the People's Republic of China | 2014CB964601 | Yixuan Wang |
| Science and Technology Commission of Shanghai Municipality | Grant 14YF1403900 | Yixuan Wang |
| Science and Technology Commission of Shanghai Municipality | Grant 15XD1503500 | Yixuan Wang |
| Ministry of Science and Technology of the People's Republic of China | 2016YFA0100400 | Shaorong Gao |
| National Natural Science Foundation of China | 31325019 | Shaorong Gao |

The funders had no role in study design, data collection and interpretation, or the decision to submit the work for publication.

### Author contributions

Yixuan Wang, Conceptualization, Validation, Investigation, Methodology, Writing—original draft, Writing—review and editing, Designed the experimental plan, derived all the cell lines, collected and analyzed data, conceived the project and provided mentoring; Chengchen Zhao, Formal analysis, Investigation, Methodology, Performed sequencing data processing, analysis and interpretation

and wrote the manuscript; Zhenzhen Hou, Formal analysis, Validation, Investigation, Visualization, Methodology, Derived all the cell lines, collected and analyzed data, and wrote the manuscript; Yuanyuan Yang, Validation, Investigation, Methodology, Sesigned the experimental plan, derived all the cell lines, collected and analyzed data, and wrote the manuscript; Yan Bi, Validation, Investigation, Methodology, Assisted with cell culture and plasmid construction; Hong Wang, Resources, Supervision, Visualization, Project administration, Provided all experimental materials; Yong Zhang, Software, Formal analysis, Methodology, Writing—original draft, Assisted with data analysis and interpretation; Shaorong Gao, Conceptualization, Resources, Supervision, Funding acquisition, Writing—original draft, Project administration, Writing—review and editing, Conceived the project and provided mentoring

### Author ORCIDs
Yixuan Wang [iD] http://orcid.org/0000-0002-2961-6760
Shaorong Gao [iD] http://orcid.org/0000-0003-1041-3928

### Ethics
Human subjects: Human subjects: Human skin specimens from abortive fetus were obtained from the Clinical and Translational Research Center of Shanghai First Maternity and Infant Hospital, Tongji University to make human embryonic fibroblasts (HEFs). The patients provided informed consent for tissue donations, and the Biological Research Ethics Committee of Tongji University approved the study.

### Decision letter and Author response
Decision letter https://doi.org/10.7554/eLife.29518.035
Author response https://doi.org/10.7554/eLife.29518.036

## Additional files

### Supplementary files
• Source data 1. Source Data for *Figures 2–6* and Figure Supplements. The Source Data files include the source R code for *Figures 2–6*. The R scripts use the data in the 'Expression Data' folder with the relative path. The 'Expression Data' folder contains the non-redundant related gene expression, methylation and histone modification tables for the main figures and figure supplements.
DOI: https://doi.org/10.7554/eLife.29518.016
• Transparent reporting form
DOI: https://doi.org/10.7554/eLife.29518.017

### Major datasets
The following dataset was generated:

| Author(s) | Year | Dataset title | Dataset URL | Database, license, and accessibility information |
|---|---|---|---|---|
| Wang Y, Zhao C, Hou Z, Yang Y, Bi Y, Wang H, Zhang Y, Gao S | 2017 | Integrative analysis of reprogramming human fibroblast cells to naïve pluripotency | https://www.ncbi.nlm.nih.gov/geo/query/acc.cgi?acc=GSE89072 | Publicly available at the NCBI Gene Expression Omnibus (accession no: GSE89072) |

The following previously published datasets were used:

| Author(s) | Year | Dataset title | Dataset URL | Database, license, and accessibility information |
|---|---|---|---|---|
| Cacchiarelli D, Trapnell C, Ziller MJ, Soumillon M, Cesana M, Karnik R, Donaghey J, Smith ZD, Ratanasirintra-woot S, Zhang X, Ho Sui SJ, Wu Z, Akopian V, Gifford CA, Doench J, Rinn JL, Daley GQ, Meissner A, Lander ES, Mikkelsen TS | 2015 | Integrative analyses of human reprogramming reveal dynamic nature of induced pluripotency | https://www.ncbi.nlm.nih.gov/geo/query/acc.cgi?acc=GSE62777 | Publicly available at the NCBI Gene Expression Omnibus (accession no: GSE62777) |
| Tang F, Qiao J, Li R | 2013 | Tracing pluripotency of human early embryos and embryonic stem cells by single cell RNA-seq | https://www.ncbi.nlm.nih.gov/geo/query/acc.cgi?acc=GSE36552 | Publicly available at the NCBI Gene Expression Omnibus (accession no: GSE36552) |
| Guo H, Zhu P, Yan L, Qiao J, Tang F | 2014 | The DNA methylation landscape of human early embryos | https://www.ncbi.nlm.nih.gov/geo/query/acc.cgi?acc=GSE49828 | Publicly available at the NCBI Gene Expression Omnibus (accession no: GSE49828) |

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
