## [Decision Letter]

Thank you for submitting your article "Unique molecular events during reprogramming of human somatic cells to induced pluripotent stem cells at naïve state" for consideration by *eLife*. Your article has been reviewed by three peer reviewers, one of whom is a member of our Board of Reviewing Editors, and the evaluation has been overseen by Fiona Watt as the Senior Editor. The following individuals involved in review of your submission have agreed to reveal their identity: Qi-Long Ying (Reviewer #2); Christine Wells (Reviewer #3).

The reviewers have discussed the reviews with one another and the Reviewing Editor has drafted this decision to help you prepare a revised submission.

Summary:

This study provides novel information on the time course of reprogramming of human fibroblasts to the naive pluripotent state.

Essential revisions:

While two reviewers agree that your study will be of interest to the field, there are some serious concerns about the comparative analysis of the gene expression data and the bioinformatics approaches used in these analyses. The reviewers also note a number of instances in which the presentation of results is unclear. You should address these major criticisms concerning data analysis and clear up the areas where the presentation lacks clarity.

Reviewer #1:

A key question relates to the analysis of the data: the authors need to clarify precisely what embryo and primed cell reprogramming datasets they are comparing their data to.

1) Subsection “Transcriptional profiling of naïve reprogramming cells” Figure 2 - how were genes classified as early embryogenesis, pre-implantation, and late embryogenesis? These do not turn up in the GO analysis in Figure 2—figure supplement 2.

2) Subsection “Transcriptional profiling of naïve reprogramming cells” - what is the significance of the activation of genes related to placenta development and gamete generation?

3) Subsection “Transcriptional profiling of naïve reprogramming cells” Figure 2 - what is the source of the human embryo data? Is there a distinction in these data for blastocyst between inner cell mass and trophectoderm? What about the epiblast proper, which the naïve cells should resemble?

4) Subsection “Transcriptional profiling of naïve reprogramming cells” – where do the authors' data show that primed cell reprogramming drops to a post-implantation stage? What post-implantation data in what species?

5) Subsection “8-cell-stage-like state was transiently established during naïve reprogramming” and elsewhere-the statement that an eight-cell stage was transiently established probably claims too much from the data. Unless I misunderstand Figure 3, the induction of a selected set of 8C genes is considerably more modest in naïve reprogrammed cells compared to that in the embryo, and the reprogramming cells may be expressing many genes that are not typical of this stage. Some other form of analysis would be required to prove that the transient cells are really equivalent to an 8C stage.

6) Subsection “8-cell-stage-like state was transiently established during naïve reprogramming” - again the significance of enrichment for genes involved in gamete generation is unclear here.

7) Subsection “Dynamic changes in epigenetic modifications in the naïve reprogramming system” and Figure 5 - what genes are chosen for study here and why?

8) Subsection “Dynamic changes in epigenetic modifications in the naïve reprogramming system” and following-the logic of comparison of the H3K4me3 with those of the mouse blastocysts is not clear. What cells of the mouse blastocyst? How about epiblast?

9) Subsection “Dynamic changes in epigenetic modifications in the naïve reprogramming system” and Figure 5 - what about global patterns of H3K9me3?

10) Discussion section – what is the basis for equating these patterns of gene expression with the epiblast? What human embryo epiblast data are used for comparison?

Reviewer #2:

Figure 2 primed heatmap is the first place that we see a comparison between the 2i reprogrammed cells and cells reprogrammed in 5iLAF. This is repeated in Figure 3. I can't find details of the primed timecourse. Was this generated in parallel with the 5iLAF 2i reprogramming? Is it a 2i timecourse or derived from the first reprogramming stage? If so, this should be included in the over-view cartoons describing the experimental design and detailed in the Materials and methods section.

Or has the data been taken from an external publication (e.g. Cacchiarelli et al., 2015 which is referenced heavily in the Results section describing the transcriptome time series)? If so, I have some concerns about the integration of these two data sets. The GEO accessions for any external data series used in comparisons should be provided in the methods. Details on normalization used to address technical batch, as would be expected from integrating disparate data sources, should be provided. These integration issues are exacerbated in comparisons of single cell RNA-seq and population RNA-seq for the correlation table with the early embryo series.

The authors benchmark the 2i 5LAF transcriptome series against an external early human embryogenesis series using a Pearson correlation. Why this choice of embryo time points? If the 2i iPSC equivalent is late epiblast, is correlation with zygote-blastocyst sufficient or informative? There are relevant later stage human and monkey embryogenesis datasets that would provide a more appropriate developmental window.

How was the correlation analysis performed? There are no substantive details of any of the bioinformatics analyses beyond the RNA-seq processing.

The comparison of 'primed'-iPSC and 5iLAF-2i-iPSC in Figure 3 to the '8-cell' signature that the authors claim is reminiscent of the reactivation of the zygote transcriptome rests heavily on comparative higher expression of this signature in the 5iLAF series and not in the 'primed' iPSC series. As far as I can tell from the methods described, only 1 replicate is provided per time point in the 5iLAF-2i-iPSC timecourse. This spike could represent an oversampling issue or some other technical issue with a small number of libraries – Panel B in Figure 6 particularly concerns us with regard to potential over-sampling. I have insufficient information to evaluate the quality of the primed and 5iLAF series without knowing the source of the former. The transposon mining comes from the same RNA-seq libraries, and so the same question about library quality, source of comparison and suspicion in a 'spike' seen in a small number of data points in a time series applies to the transposon analysis.

The authors have not evaluated whether the 8-cell signature is expressed later in human embryonic development. Indeed, many of the individual genes are more broadly expressed. This needs further exploration before concluding that the 5iLAF-iPSC have an equivalent zygote-reactivation wave of chromatin remodeling and gene expression. Of the individual genes highlighted in the RNA-seq, and then confirmed by IF, Figure 3 lacks comparison of the 5iLAF-iPSC with the primed iPSC series or with other pluripotent cell lines, so the claim that these genes are hall-marks of the naïve state remains somewhat anecdotal.

[Editors' note: further revisions were requested prior to acceptance, as described below.]

Thank you for submitting your article "Unique molecular events during reprogramming of human somatic cells to induced pluripotent stem cells (iPSCs) at naïve state" for consideration by *eLife*. Your article has been re-reviewed by one peer reviewer, and the evaluation has been overseen by a Reviewing Editor and Fiona Watt as the Senior Editor. The following individual involved in review of your submission has agreed to reveal her identity: Christine Wells (Reviewer #3).

The reviewers have discussed the reviews with one another and the Reviewing Editor has drafted this decision to help you prepare a revised submission.

Summary:

The revised manuscript and rebuttal address many concerns raised in the first round of review. However, one reviewer who is highly experienced in this field continues to have several important concerns about the analysis and interpretation of the data. These are summarized below and relate to your analysis of the repeat RNA-seq time course, the specificity of expression of key markers used to define cell state and the reliance on one embryo dataset only, and the degree to which the final cell state of the naïve and primed protocols really differ.

We are therefore offering you the opportunity to address these issues. As a matter of policy, we rarely offer more than one round of revision so please take the following concerns carefully as we may not be able to consider any further revisions.

Essential revisions:

1) The reanalysis, including repeat RNA-seq of the time course, have been superficially addressed. In fact, it is not clear that this repeat data has been incorporated into any of the analyses or figures. The libraries appear genuinely different, sufficiently so that the numbers of DE genes, and scales on plots should be different between the manuscript versions. They are not. One would assume that this would address potential library artefacts association with the earlier version of the manuscript, however no changes in relevant figures have arisen from inclusion of additional data, suggesting that no additional data has been included in the analysis.

Please clarify whether or not the repeat time course has been incorporated into the revised figures.

2) Figure 2 I'm concerned that genes I observe to be highly variable in any pluripotent cell culture, including 2i-cultures, are being used as hallmarks of naïve vs primed states, especially given the shallow replication of the data generated here, and the single comparison to one primed dataset generated in another lab. In particular, the claims that dox-withdrawal in primed iPSC (Cacchiarelli) lead to loss of naïve markers such as DPP2,3,5, DNMT3L, NODAL etc. exemplifies the superficial nature of the comparison being made in this manuscript. These are genes that are highly variable in hESC and iPSC cultures, regardless of reprogramming method or 2i-culture condition (see for example https://www.stemformatics.org/expressions/result?graphType=box&datasetID=7253&gene=ENSG00000156574&db_id=56 as well as https://www.stemformatics.org/expressions/result?graphType=box&datasetID=7045&gene=ENSG00000187569&db_id=56). There is not sufficient depth of characterization in this manuscript to demonstrate that gain or loss of these markers, in either reprogramming study (present or Cacchiarelli), are indicative of a higher-order difference in primed or naïve state. The authors can verify the variable expression of markers in the HipSc catalogue, and in expression atlases such as GEO, or stemformatics, which include a large number of iPSC reprogramming datasets, including datasets linked above.

The authors have not adequately addressed the question of comparison to late epiblast stages in their correlation analyses – the Yan dataset is small, and Gao et al., seem to be using hESC as their equivalent to a post-implantation stage. So my original request for more appropriate embryo comparisons have not been addressed. The Yan dataset used post-hoc clustering to predict which cells were PE vs EPI vs Polar TE. The correlations are interesting observations but overstated because they are looking at a limited range of embryonic stages.

Please address the issue of variable expression of markers used to define cell states and refer to other studies of human embryo gene expression to validate markers for particular cell states.

3) The behaviour of primed and naive time courses are very similar up to the derivation of the final iPSC line. I'm not convinced that differences in the final iPSC state are not the result of normal line variation.

Please highlight more clearly the basis for claiming that the end stages of these protocols are genuinely different rather than a function of normal variation between cell lines. Is it possible for instance that the differences in the final cell lines reflect long term adaptation to particular culture conditions rather than inherent differences in cell state? If 2 above is answered adequately, you should be able to address this point. We apologise if the significance of point 2 above was not clear enough in the previous review.

It is important to address these concerns by demonstrating (1) that your analysis of the repeat data is adequately reflected in the revision, (2) by showing that the markers used clearly define what they are claimed to (by reference to databases or more recent human embryo data), and (3) to define clearly the differences in the reprogramming endpoints. These revisions do not entail new experimentation and should not consume too much time.

---

## [Author Response]

Reviewer #1:A key question relates to the analysis of the data: the authors need to clarify precisely what embryo and primed cell reprogramming datasets they are comparing their data to.

Thank the reviewer for the question. Both the embryo development data and primed cell reprogramming data are published datasets. The embryo development dataset is GSE36552 (Yan et al., 2013). The primed cell reprogramming dataset is GSE62777 (Cacchiarelli et al., 2015). We have added these statements to the revised manuscript.

1) Subsection “Transcriptional profiling of naïve reprogramming cells” Figure 2 - how were genes classified as early embryogenesis, pre-implantation, and late embryogenesis? These do not turn up in the GO analysis in Figure 2—figure supplement 2.

Thank the reviewer for the question. The genes in Figure 2 were classified according to the classification criteria established in Cacchiarelli et al., 2015, in which they applied gene ontology enrichment analysis for both biological processes and developmental cell identity (Edgar et al., (2013)). We could also observe signature genes (cited in Figure 2—figure supplement 2) representative for each cluster in our system as well as in Cacchiarelli et al., 2015. We also added these statements to the Materials and methods section.

2) Subsection “Transcriptional profiling of naïve reprogramming cells” - what is the significance of the activation of genes related to placenta development and gamete generation?

Thank the reviewer for the question. According to reviewer #2’s suggestion, we performed RNA-seq of another replicate during the naïve reprogramming time-course and re-analyzed the datasets after incorporating some samples of the new replicate. The GO enrichment analysis was also re-performed (Figure 2—figure supplement 2), and there were some small changes in the GO enrichment terms. The p-value of these genes related to gamete generation is 0.001934. And we have added that in the revised manuscript.

3) Subsection “Transcriptional profiling of naïve reprogramming cells” Figure 2-what is the source of the human embryo data? Is there a distinction in these data for blastocyst between inner cell mass and trophectoderm? What about the epiblast proper, which the naïve cells should resemble?

Thank the reviewer for the question. The source of the human embryo dataset is GSE36552 (Yan et al., 2013). In their paper, they distinguished the cells at the late blastocyst stage into Mural trophectoderm (TE), Polar TE, epiblast (EPI), and primitive endoderm (PE) by unsupervised clustering of the expression profiles in each cell. They also derived hESCs from the blastocyst and performed RNA-seq analysis of hESC0 (passage 0) and hESC10 (passage 10) to simulate the in vivo post-implantation stages during development. This dataset includes pre-implantation, implantation and post-implantation stages during development. We re-compared the correlation of gene expression between cells during naïve reprogramming process and embryonic cells during development, and found that the reprogramming cells at day 20 and 24 and niPSC-Ts most closely resembled human embryos at the late blastocyst stage, which represents the time window of naïve ground state in vivo (Hackett and Surani, 2014; Nichols and Smith, 2009) We have updated the Figure 2 and the revised manuscript.

4) Subsection “Transcriptional profiling of naïve reprogramming cells” – where do the authors' data show that primed cell reprogramming drops to a post-implantation stage? What post-implantation data in what species?

Thank the reviewer for the question. The human primed cell reprogramming dataset is GSE62777 (Cacchiarelli et al., 2015). In this paper, the authors claimed that the primed reprogramming cells show transient re-activation of developmental genes, eventually reaching a pre-implantation-like state that is lost upon derivation of primed iPSC lines.

By comparing the correlation of gene expression between primed cell reprogramming dataset (GSE62777) and embryo development dataset (GSE36552), we could also find that the cells at 24d+dox of primed reprogramming most closely resembled to the epiblast (EPI) stage during embryonic development. After dox withdrawal, the cells most closely resembled to the hESC0 and hESC10 cells, which represented the post-implantation stage (Author response image 1).

5) Subsection “8-cell-stage-like state was transiently established during naïve reprogramming” and elsewhere-the statement that an eight-cell stage was transiently established probably claims too much from the data. Unless I misunderstand Figure 3, the induction of a selected set of 8C genes is considerably more modest in naïve reprogrammed cells compared to that in the embryo, and the reprogramming cells may be expressing many genes that are not typical of this stage. Some other form of analysis would be required to prove that the transient cells are really equivalent to an 8C stage.

Thank the reviewer for this important question. The statement that transcriptional network with eight-cell-stage-like characteristics was transiently activated during naïve reprograming was concluded based on the following observations:

1) 8-cell specific genes were significantly enriched in the cluster exhibiting transiently enhanced transcription activity during the late stages of naïve reprogramming, which could not be observed during primed reprogramming process.

2) Dissecting the dynamics of transposon elements (TEs) by similar strategy also revealed that 8C-TEs were transiently re-activated during naïve reprograming.

3) The H3K9me3 signals around the 8C-TEs exhibited a transiently decreased pattern at day 14, strongly correlated with the transiently increasing transcriptional activity of the 8C-TEs at that time point.

Our results cannot reach the conclusion that the transient cells are equivalent to an 8C stage, and we apologize for the misleading. We would like to conclude that the specific pattern of 8C-transcripts wave that we observed suggests the emergence of gene network activation with the 8C-stage-like characteristics during naïve reprograming.

6) Subsection “8-cell-stage-like state was transiently established during naïve reprogramming”-again the significance of enrichment for genes involved in gamete generation is unclear here.

The p-value of these genes related to gamete generation is 3.784322e^-08^. We have added that in the revised manuscript.

7) Subsection “Dynamic changes in epigenetic modifications in the naïve reprogramming system” and Figure 5 - what genes are chosen for study here and why?

Thank the reviewer for the question. The genes with both H3K4me3 and H3K27me3 modifications covered at least one time point in the promoter regions (+/- 2k of TSS) during the reprogramming process were chosen for our study. The genes with no H3K4me3 and H3K27me3 modifications in the promoter regions are excluded since they are not relevant for the bivalency analysis.

8) Subsection “Dynamic changes in epigenetic modifications in the naïve reprogramming system” and following-the logic of comparison of the H3K4me3 with those of the mouse blastocysts is not clear. What cells of the mouse blastocyst? How about epiblast?

The reviewer raised a very good point. We hoped to compare the epigenetic similarity between the reprograming cells and the embryonic cells during development. Since there is no H3K4me3 ChIP-seq data in human pre-implantation embryos, we used the mouse pre-implantation embryo H3K4me3 ChIP-seq dataset (GSE73952) and calculated the correlation of H3K4me3 signal in the above two systems at the promoter regions (+/- 2k of TSS) of homolog genes.

Thank the reviewer for pointing this. We now realize that this analysis may be inaccurate and inappropriate. We have removed the figure panel and the related statements in our revised manuscript.

9) Subsection “Dynamic changes in epigenetic modifications in the naïve reprogramming system” and figure 5EF-what about global patterns of H3K9me3?

The global pattern of H3K9me3 is down-regulated, as shown in Author response image 2:

**Author response image 2. respfig2:** 

10) Discussion section – what is the basis for equating these patterns of gene expression with the epiblast? What human embryo epiblast data are used for comparison?

Thank the reviewer for the question. We apologize for the misleading caused by the inaccurate statements. Our analysis showed that derived niPSC-Ts exhibit reactivation of network with late blastocyst-stage signatures. We removed such statements for more accuracy in the revised manuscript.

Reviewer #2:Figure 2 primed heatmap is the first place that we see a comparison between the 2i reprogrammed cells and cells reprogrammed in 5iLAF. This is repeated in Figure 3. I can't find details of the primed timecourse. Was this generated in parallel with the 5iLAF 2i reprogramming? Is it a 2i timecourse or derived from the first reprogramming stage? If so, this should be included in the over-view cartoons describing the experimental design and detailed in the Materials and methods section.Or has the data been taken from an external publication (e.g. Cacchiarelli et al., 2015 which is referenced heavily in the Results section describing the transcriptome time series)? If so, I have some concerns about the integration of these two data sets. The GEO accessions for any external data series used in comparisons should be provided in the methods. Details on normalization used to address technical batch, as would be expected from integrating disparate data sources, should be provided. These integration issues are excerbated in comparisons of single cell RNA-seq and population RNA-seq for the correlation table with the early embryo series.

Thank the reviewer for this important question. The primed reprogramming dataset is taken from the publication “Cacchiarelli et al., 2015”, and the GEO accession number is GSE62777. We apologize for our omissions in the description. We have added the GEO accessions for both primed reprogramming and early embryo development data series in the Materials and methods section. Batch effects of samples from different systems are removed using removeBatchEffect function of the R library “edgeR”. The negative values in the normalized data are considered as zero. We have also provided the details on normalization used to address technical batch between different systems in the revised Materials and methods section.

The authors benchmark the 2i 5LAF transcriptome series against an external early human embryogenesis series using a Pearson correlation. Why this choice of embryo time points? If the 2i iPSC equivalent is late epiblast, is correlation with zygote-blastocyst sufficient or informative? There are relevant later stage human and monkey embryogenesis datasets that would provide a more appropriate developmental window.

Thank the reviewer for the important question. The source of the human embryo dataset is dataset is GSE36552 (Yan et al., 2013). In this paper, they distinguished the cells at the late blastocyst stage into Mural trophectoderm (TE), Polar TE, epiblast (EPI), and primitive endoderm (PE) by unsupervised clustering of the expression profiles in each cell. They also derived hESCs from the blastocyst and performed RNA-seq analysis of hESC0 (passage 0) and hESC10 (passage 10) to simulate the in vivo post-implantation stages during development. This dataset includes pre-implantation, implantation and post-implantation stages during development.

We re-compared the correlation of gene expression between cells during naïve reprogramming process and embryonic cells during development (Figure 2) and found that the reprogramming cells at day 20 and 24 and niPSC-Ts most closely resembled human embryos at the late blastocyst stage, which represents the time window of naïve ground state in vivo (Hackett, and Surani, 2014; Nichols and Smith, 2009)

How was the correlation analysis performed? There are no substantive details of any of the bioinformatics analyses beyond the RNA-seq processing.

The Pearson correlation coefficient was calculated between each two samples on common genes using R function cor(). We apologize for our omissions in the method description. The substantive details of bioinformatics analyses are displayed below:

Gene expression Analysis

Batch effects of samples from different systems are removed using *removeBatchEffect* function of the R library “edgeR”. The negative values in the normalized data are considered as zero.

Retrotransposon expression Analysis

A time-point specific expressed score was calculated as the expression of given time point divided by the average expression for each candidate gene, which should have a max expressed value in the given stage. The top 100 8C-TEs are the top 100 TE ranked by the time-point specific expressed score.

Gene ontology analysis

Functional annotation was performed using the Database for Annotation, Visualization and Integrated Discovery (DAVID) Bioinformatics Resource (Huang et al., 2009). Gene ontology terms for each function cluster were summarized to a representative term and P-values were plotted to show the significance. The genes are classified according to developmental cell identity using LifeMap Discovery (Edgar et al., 2013).

And we replenished these bioinformatics analysis methods to “Materials and methods” section.

The comparison of 'primed'-iPSC and 5iLAF-2i-iPSC in Figure 3 to the '8-cell' signature that the authors claim is reminiscent of the reactivation of the zygote transcriptome rests heavily on comparative higher expression of this signature in the 5iLAF series and not in the 'primed' iPSC series. As far as I can tell from the methods described, only 1 replicate is provided per time point in the 5iLAF-2i-iPSC timecourse. This spike could represent an oversampling issue or some other technical issue with a small number of libraries – Panel B in Figure 6 particularly concerns us with regard to potential over-sampling. I have insufficient information to evaluate the quality of the primed and 5iLAF series without knowing the source of the former. The transposon mining comes from the same RNA-seq libraries, and so the same question about library quality, source of comparison and suspicion in a 'spike' seen in a small number of data points in a time series applies to the transposon analysis.

Thank the reviewer for this important question. The primed reprogramming dataset is taken from the publication “Cacchiarelli et al., 2015”, and the GEO accession number is GSE62777.

According to the reviewer’s suggestion, we performed another replicate per time point in the naïve reprogramming timecourse, and chose the qualified samples at 8d, 14d and 20d during naïve reprogramming, which were also important time points for naïve reprogramming. Then we re-analyzed the datasets and got the similar conclusions with small variations (Figure 2, Figure 2—figure supplement 1 and Figure 2—figure supplement 2, Figure 3, Figure 3—figure supplement 1 and Figure 3—figure supplement 2, Figure 6, Figure 6—figure supplement 1).

The authors have not evaluated whether the 8-cell signature is expressed later in human embryonic development. Indeed, many of the individual genes are more broadly expressed. This needs further exploration before concluding that the 5iLAF-iPSC have an equivalent zygote-reactivation wave of chromatin remodeling and gene expression. Of the individual genes highlighted in the RNA-seq, and then confirmed by IF, Figure 3 lacks comparison of the 5iLAF-iPSC with the primed iPSC series or with other pluripotent cell lines, so the claim that these genes are hall-marks of the naïve state remains somewhat anecdotal.

Thank the reviewer for this important question. We have now evaluated the expression of 8C-genes in later stages including the PGC cells and somatic cells at 4 weeks, 7weeks and 8weeks (GSE63818) during human embryonic development (Guo et al., 2015), which showed no significant expression compared to the 8-cell stage (Author response image 3).

**Author response image 3. respfig3:** 

For the individual genes highlighted in Figure 3 such as MBD3L2-5 and ZSCAN4, we also checked their expression dynamics during primed reprogramming. Both IF and western blot analysis exhibited no significant activation of these genes during the primed reprogramming process. We have also added the immunostaining and western blotting results of primed series to the revised Figure 3 and Figure 3—figure supplement 1.

[Editors' note: further revisions were requested prior to acceptance, as described below.]

Essential revisions:1) The reanalysis, including repeat RNA-seq of the time course, have been superficially addressed. In fact, it is not clear that this repeat data has been incorporated into any of the analyses or figures. The libraries appear genuinely different, sufficiently so that the numbers of DE genes, and scales on plots should be different between the manuscript versions. They are not. One would assume that this would address potential library artefacts association with the earlier version of the manuscript, however no changes in relevant figures have arisen from inclusion of additional data, suggesting that no additional data has been included in the analysis.Please clarify whether or not the repeat time course has been incorporated into the revised figures.

Thank the reviewer for this question. The third replicate of RNA-seq datasets of the naïve reprogramming time course has been analyzed and incorporated into the figures (Figure 2, Figure 2—figure supplement 1; Figure 2—figure supplement 2; Figure 3; Figure 3—figure supplement 1; Figure 3—figure supplement 2; Figure 6). And the numbers of DE genes in each cluster (Figure 2—figure supplement 2) were different from the first submission version in our last revision. We apologize for our omission in the legend of Figure 2 and in the update of Figure 2—figure supplement 1, Figure 3, Figure 3, Figure 3—figure supplement 1, Figure 3—figure supplement 2. We have updated these figures in this revised version.

2) Figure 2'm concerned that genes I observe to be highly variable in any pluripotent cell culture, including 2i-cultures, are being used as hallmarks of naïve vs primed states, especially given the shallow replication of the data generated here, and the single comparison to one primed dataset generated in another lab. In particular, the claims that dox-withdrawal in primed iPSC (Cacchiarelli) lead to loss of naïve markers such as DPP2,3,5, DNMT3L, NODAL etc. exemplifies the superficial nature of the comparison being made in this manuscript. These are genes that are highly variable in hESC and iPSC cultures, regardless of reprogramming method or 2i-culture condition (see for example https://www.stemformatics.org/expressions/result?graphType=box&datasetID=7253&gene=ENSG00000156574&db_id=56 as well as https://www.stemformatics.org/expressions/result?graphType=box&datasetID=7045&gene=ENSG00000187569&db_id=56. There is not sufficient depth of characterization in this manuscript to demonstrate that gain or loss of these markers, in either reprogramming study (present or Cacchiarelli), are indicative of a higher-order difference in primed or naïve state. The authors can verify the variable expression of markers in the HipSc catalogue, and in expression atlases such as GEO, or stemformatics, which include a large number of iPSC reprogramming datasets, including datasets linked above.

Thank the reviewer for this important question. We have now re-selected the marker genes that indicate naïve pluripotency according to several important publications in the naïve pluripotency research field including Theunissen et al., (2014), Takashima et al., (2014), Pastor et al., (2016), and Guo et al., (2016). We also collected the published RNA-seq datasets containing 5 sets of naïve and primed ESCs and 3 sets of naïve and primed iPSCs to validate the selected marker genes expression according to the reviewer’s suggestion (GSE59435 of Theunissen et al., paper; E-MTAB-2857 of Takashima et al., paper; GSE76970 of Pastor et al., paper; GSE69319 of Yang et al., paper). We compared the expression of these marker genes in these datasets. Although some markers exhibit variable expression levels in different cell lines, most of them showed significant expression difference between the naïve cells and primed cells (Author response image 4), t-test are performed on each available pair of datasets, p-value are shown in the right of the plots (NA stands for not enough data for t-test)).

**Author response image 4. respfig4:** 

Moreover, we also checked the expression dynamics of the re-selected markers in the iPSC reprogramming datasets (GSE50206 from Takahashi K et al., (2014)), most of which also showed transiently upregulation during the reprogramming process, and finally down-regulated in the iPSC stage (Author response image 5).

**Author response image 5. respfig5:** 

The authors have not adequately addressed the question of comparison to late epiblast stages in their correlation analyses – the Yan dataset is small, and Gao et al., seem to be using hESC as their equivalent to a post-implantation stage. So my original request for more appropriate embryo comparisons have not been addressed. The Yan dataset used post-hoc clustering to predict which cells were PE vs EPI vs Polar TE. The correlations are interesting observations but overstated because they are looking at a limited range of embryonic stages.Please address the issue of variable expression of markers used to define cell states and refer to other studies of human embryo gene expression to validate markers for particular cell states.

Thank the reviewer for this question. According to the reviewer’s suggestion, we have added two more human embryonic development datasets, including RNA-seq dataset of single embryonic cells from E3 to E7 (ArrayExpress: E-MTAB-3929) that used PAM clustering to classified which cells were PE *vs* EPI *vs* PTE, and RNA-seq datasets of somatic cells at 4 weeks, 7 weeks and 8 weeks (GSE63818) during human embryonic development (Guo et al., 2015), which is the earliest embryonic datasets after implantation to our knowledge. The correlation analyses between naïve reprogramming dataset and human E3-E7 datasets also showed that the reprogramming cells at day 20 and 24 and niPSC-Ts most closely resembled epiblast cells at E5 to E7, especially at E6, similar to our previous observations in the correlation analyses between the naïve reprogramming dataset and Yan et al.’s dataset (GSE36552) (Author response image 6). However, this strong correlation was not observed when comparing our naïve reprogramming dataset with Guo et al., dataset (GSE63818) (Author response image 7).

**Author response image 6. respfig6:** 

**Author response image 7. respfig7:** 

We are sorry that we have not found the public embryonic data at the stages just in the narrow time window of post-implantation. Therefore, in the last revised response we used hESC0 and hESC10 to simulate the post-implantation time-point during development, which were reported to originate from a post-ICM intermediate (PICMI), a transient epiblast-like structure that has undergone X-inactivation in female cells and is both necessary and sufficient for ESC derivation (O’Leary et al., 2012).

3) The behaviour of primed and naive time courses are very similar up to the derivation of the final iPSC line. I'm not convinced that differences in the final iPSC state are not the result of normal line variation.Please highlight more clearly the basis for claiming that the end stages of these protocols are genuinely different rather than a function of normal variation between cell lines. Is it possible for instance that the differences in the final cell lines reflect long term adaptation to particular culture conditions rather than inherent differences in cell state? If 2 above is answered adequately, you should be able to address this point. We apologise if the significance of point 2 above was not clear enough in the previous review.

Thank the reviewer for this important question. We have performed cell clustering based on cell-to-cell correlation of the reprogramming endpoints in our study and naïve/primed ESCs/iPSCs from public datasets mentioned above. The correlation analyses showed that the niPSC-Ts which most resembled other naïve ESCs/iPSCs in the public datasets, separate clearly from the piPSC-Ts, indicating that the differences in the final iPSC state are not the result of normal line variation (Author response image 8).

We do not think that the differences between niPSC-T and piPSC-Ts reflect long term adaptation to the 5iLAF culture conditions. The activation of OCT4 distal enhancer and the signatures in epigenome in naïve cells strongly suggest their inherent differences from primed cells.

**Author response image 8. respfig8:**